# What Makes Better Augmentation Strategies? Augment Difficult but Not too Different

**Jaehyung Kim[1], Dongyeop Kang[2], Sungsoo Ahn[3], Jinwoo Shin[1]**
[1] Korea Advanced Institute of Science and Technology (KAIST)
[2] University of Minnesota (UMN) [3] Pohang University of Science and Technology (POSTECH)
{jaehyungkim, jinwoos}@kaist.ac.kr, dongyeop@umn.edu
sungsoo.ahn@postech.ac.kr

## Abstract

The practice of data augmentation has been extensively used to boost the performance of deep neural networks for various NLP tasks. It is more effective when only a limited number of labeled samples is available, *e.g.*, low-data or class-imbalanced regimes. Most current augmentation techniques rely on parameter tuning or inherent randomness; hence, their effectiveness largely varies on the tasks. To efficiently find the best augmentation strategy for each task, learning data augmentation policy is a promising solution, but the question of what makes a good augmentation in NLP tasks and how to design the reward function for learning a good policy remains under-explored. To answer this, we hypothesize that good data augmentation should construct more diverse and challenging samples for providing informative training signals, while avoiding the risk of losing the semantics of original samples. Therefore, we design a novel reward function for updating the augmentation policy to construct *difficult* but *not too different* samples (DND). Particularly, we jointly optimize a data augmentation policy while training the model, to construct the augmented samples with low confidence but a high semantic similarity with original ones. In addition, we introduce a sample re-weighting scheme to focus on difficult augmented samples after the original ones are learned confidently for more effective learning from the augmented ones. Our learning-based augmentation outperforms the recent state-of-the-art augmentation schemes on various text classification tasks and GLUE benchmark by successfully discovering the effective augmentations for each task. Remarkably, our method is more effective on the challenging low-data and class-imbalanced regimes, and the learned augmentation policy is well-transferable to the different tasks and models.

## 1 Introduction

Deep neural networks (DNNs) have shown near human- or superhuman-level performances on various NLP benchmark tasks. This success, however, crucially relies on the availability of large labeled datasets, which typically require a lot of human efforts to be constructed (Brown et al., 2020; Cheng et al., 2020). Although recent advance in language models (LMs) significantly boosts state-of-the-art performances using self-supervised pre-training with a massive unlabeled dataset (Devlin et al., 2018; Liu et al., 2019), the number of given labeled samples on each downstream task is still critical for the performance and stability of fine-tuned LMs (Sun et al., 2019; Zhang et al., 2020b). Data augmentation is one of the most effective ways to efficiently use the given labeled samples by enlarging the amount and diversity with label-preserving transformations, so it helps to improve the generalization of DNNs (Lim et al., 2019). Hence, the practice of data augmentation has been extensively used to boost the performance of DNNs and LMs for various NLP tasks (Jiang et al., 2020; Qu et al., 2021), and it is more effective when only a limited number of labeled samples is available, *e.g.*, low-data or class-imbalanced regimes (Xie et al., 2020; Kim et al., 2020).

Most current augmentation techniques rely on parameter tuning (Zhu et al., 2020) or inherent randomness (Wei & Zou, 2019); hence, their effectiveness largely varies between the tasks (see Figure 1(a)). To efficiently find the best augmentation strategy for each task, learning augmentation policy (*i.e.,* distribution of multiple augmentations) has been explored as a promising solution (Cubuk

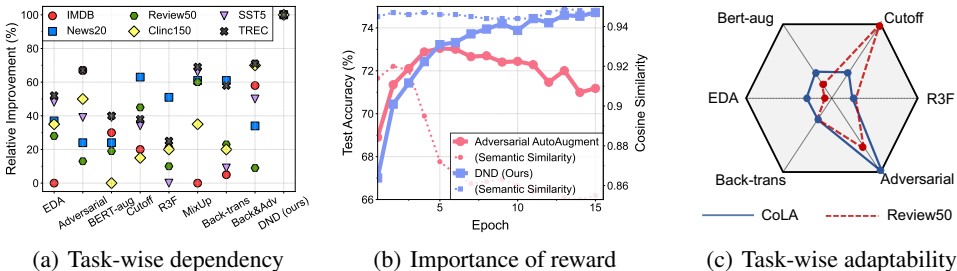

(a) Task-wise dependency  (b) Importance of reward  (c) Task-wise adaptability

Figure 1: (a) Task-dependency of each augmentation method measured on 5-shot tasks in Section 4.2. (b) Test accuracy and semantic similarity between original and augmented samples under different reward functions to optimize the augmentation policy: Adversarial AutoAugment (Zhang et al., 2020c) and DND (Ours). (c) Distribution of augmentations learned via DND. Depend on a given task, the distribution of augmentations is automatically adapted.

et al., 2019; Hataya et al., 2020). Here, a key for successful policy learning is on the design of reward function used to update it, which implicates what a good augmentation is; hence, various rewards are studied especially under vision tasks, *e.g.,* maximizing validation accuracy (Cubuk et al., 2019) or training loss (Zhang et al., 2020c). However, we found that these existing reward functions would not be enough for learning effective augmentation policy in NLP tasks. This is due to the different nature of NLP tasks from discrete input space or vulnerability of the semantics for a slight modification (Wang et al., 2018; Garg & Ramakrishnan, 2020). For example, we found that the recent state-of-the-art augmentation learning approach used for image classification is not effective for NLP tasks as it suffers from losing the semantics of original sentence in augmentations (see Figure 1(b)). This motivates us to explore what criteria make a good augmentation, and how to design the reward function for policy to find them in NLP tasks.

**Contribution.** In this paper, we develop a simple yet effective augmentation scheme for NLP tasks, coined learning to augment **D**ifficult, but **N**ot too **D**ifferent (DND). We first design a novel reward function for updating augmentation policy under the following intuition: an effective augmented sample should be more *difficult* than the original sample for providing 'informative' training signals, while maintaining its semantic meaning *not too different* from the original for avoiding 'wrong' signals. Specifically, we compose the reward function using two complementary ingredients from training model: (1) model's training loss for the given downstream task (on augmented samples) and (2) semantic similarity (between augmented and original samples) measured by using the contextualized sentence embeddings from the training language model.

We further apply a sample-wise re-weighting scheme when updating the model and policy with the augmented samples, to learn them more effectively by incorporating the original sample's learning status. Specifically, we assign more weight on the augmented sample, which (1) its corresponding original sample is learned enough, *i.e.*, has high confidence, and (2) the confidence gap between original and augmented samples is large. With the proposed reward function and re-weighting scheme, the augmentation policy is simultaneously optimized during the model fine-tuning with a gradient-based optimization, by applying an efficient continuous relaxation to the non-differentiable components within the augmentation policy (*e.g.,* sampling the augmentations from the policy).

We demonstrate the effectiveness of the proposed augmentation policy learning scheme on various text classification datasets and GLUE benchmark (Wang et al., 2019), where our method consistently improves over the recent state-of-the-art augmentation schemes by successfully discovering the effective augmentation methods for each task (see Figure 1(c)). For example, on the six different text classification datasets, DND exhibited 16.45% and 8.59% relative test error reduction on average, compared to the vanilla and the previous best augmentation method, respectively. We also found that DND is more effective on the challenging low-resource and class-imbalanced regimes, and the learned augmentation policy can be easily transferable to the different tasks and models. This implies the broad applicability of our augmentation scheme: for instance, it might substitute the cost of augmentation learning on new tasks by using the pre-trained augmentation policies from DND.

## 2  RELATED WORKS

**Data augmentation in NLP tasks.** Recently, the interest in data augmentation has also increased on NLP tasks (Feng et al., 2021) and various data augmentation approaches have been proposed,

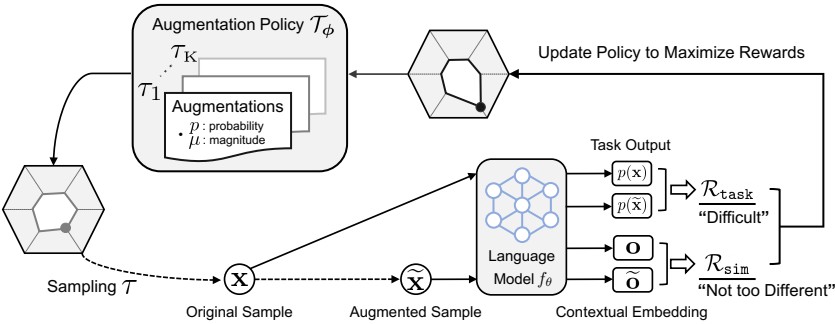

Figure 2: Illustration of learning to augment **D**ifficult, but **N**ot too **D**ifferent (DND).

such as word replacement with pre-defined rules (Kang et al., 2018; Wei & Zou, 2019) or pre-trained language models (Yi et al., 2021), back-translation (Xie et al., 2020), MixUp (Guo et al., 2019), and Cutoff (Shen et al., 2020). In a broader sense, adversarial training (Jiang et al., 2020; Zhu et al., 2020) can also be viewed as constructing augmented samples by adding perturbations to the word embedding. Stacking of these augmentations has been shown to provide further improvement (Qu et al., 2021), but it requires the exhaustive searching cost for exploring each configuration one by one. Hence, the analysis what makes a good data augmentation is an emerging research direction (Miyato et al., 2018; Gontijo-Lopes et al., 2020). For example, after the extensive study on the existing augmentations in the vision tasks, Gontijo, et al. (Gontijo-Lopes et al., 2020) identify that the successful augmentations have the following properties: higher training loss and smaller distribution shift from the original. But, any specific way to obtain such a desired augmentation is not suggested.

**Learning-based data augmentation.** Automatically finding the effective data augmentation from data has naturally emerged to overcome the exhausting search cost from deciding which augmentation would be applied and tuning its parameters. Learning augmentation policy (*i.e.,* distribution of multiple augmentations), optimized to find the effective augmentation given a huge search space, is the representative method for this. Here, a key for successful policy learning is a design of reward function used to update it; hence, various objectives for learning augmentation policy have been proposed, especially in the computer vision. Maximizing the validation accuracy is arguably one of the most natural and successful approaches (Cubuk et al., 2019; Hu et al., 2019). Also, matching the distributions of original and augmented samples is popularly used as a learning objective by considering data augmentation as the problem of filling the missing data points of training distribution (Tran et al., 2017; Hataya et al., 2020). Recently, the augmentation policy optimized to increase the training loss outperforms previous approaches (Zhang et al., 2020c; Wu et al., 2020).

On the other hand, finding the efficient optimization method for learning augmentation policy is also widely explored to overcome the non-differentiability within the augmentation policy (*e.g.,* sampling the augmentations from the policy). Reinforcement learning has been the representative method (Cubuk et al., 2019; Zhang et al., 2020c), but more efficient continuous relaxation has been recently proposed (Hataya et al., 2020; Li et al., 2020). However, such learning-based data augmentation schemes have been under-explored for NLP tasks (Ren et al., 2021).

## 3 DND: AUGMENT DIFFICULT, BUT NOT TOO DIFFERENT

### 3.1 OVERVIEW AND PROBLEM DESCRIPTION

**Overview.** In this section, we present our technique, coined learning to augment Difficult, but Not too Different (DND). Our main idea is optimizing the augmentation policy to construct a sample that (a) is more difficult than the original sample for improving the generalization of the classifier, yet (b) maintains the semantics of the original sample. In Section 3.2, we elaborate the reward and the training objective for our augmentation policy.[1]

**Problem description.** We first describe the problem setup of our interest under a text classification scenario. Let $\mathcal{D}$ denote the given training dataset consisting of tuples $(\mathbf{x}, y) \in \mathcal{D}$ where $\mathbf{x} =$

---

[1]We use the terminologies of policy and reward following previous works (Li et al., 2020; Hataya et al., 2020), although our approach is not exactly a reinforcement learning algorithm.

$[x_1, \ldots, x_L]$ and $y$ are the sequence of input tokens and the target label, respectively. Our goal is to train a classifier $f_\theta(\mathbf{x})$, initialized with a pre-trained transformer-based language model (*e.g.,* BERT (Devlin et al., 2018)), to minimize the task-specific loss $\mathcal{L}_{\texttt{task}}(\mathbf{x}, y)$ such as a cross-entropy loss $\ell_{\texttt{xe}}(p(\mathbf{x}), y)$ where $p(\mathbf{x}) = \text{Softmax}\big(f_\theta(\mathbf{x})\big)$. For improving the model's generalization upon a (limited) training data $\mathcal{D}$, one can construct the augmented sample $\widetilde{\mathbf{x}} = \tau(\mathbf{x})$ by applying a label-preserving transformation $\tau$. In contrast to existing works relying on parameter tuning or inherent randomness for finding effective $\tau$, we consider optimizing the augmentation policy $\mathcal{T}_\phi$ where $\tau \sim \mathcal{T}_\phi$ to maximize a reward function $\mathcal{R}$.

## 3.2 Reward and Training Objective

To train our data augmentation policy $\mathcal{T}_\phi$, we train it to maximize the following reward function:
$$\mathcal{R}(\mathbf{x}, \widetilde{\mathbf{x}}, y) = \mathcal{R}_{\texttt{task}}(\widetilde{\mathbf{x}}, y) + \lambda_{\texttt{s}}\mathcal{R}_{\texttt{sim}}(\mathbf{x}, \widetilde{\mathbf{x}}),$$
where $\widetilde{\mathbf{x}} = \tau(\mathbf{x})$, $\tau \sim \mathcal{T}_\phi$. $\mathcal{R}_{\texttt{task}}$ and $\mathcal{R}_{\texttt{sim}}$ are the newly proposed reward functions to (1) provide 'informative' learning signal to training classifier and (2) keep the semantic meaning of augmented sample $\widetilde{\mathbf{x}}$ to be similar to the original $\mathbf{x}$, respectively.

**Rewarding difficult samples.** To provide informative training signals, we encourage the augment policy to construct a harder view of the original sample; hard samples are known to facilitate training for many tasks, *e.g.*, object detection (Shrivastava et al., 2016), curriculum learning (Bengio et al., 2009) and data augmentation (Zhang et al., 2020c; Wu et al., 2020). To this end, we reward the policy for generating an augmented sample $\widetilde{\mathbf{x}} = \tau(\mathbf{x})$, $\tau \sim \mathcal{T}_\phi$ with high training loss, while down-weighing the case where the high training loss incurred from the original sample $\mathbf{x}$, not the augmentation policy $\mathcal{T}_\phi$. To this end, we update the augmentation policy $\mathcal{T}_\phi$ to maximize the following reward:
$$\mathcal{R}_{\texttt{task}}(\widetilde{\mathbf{x}}, y) = w(\widetilde{\mathbf{x}})\mathcal{L}_{\texttt{task}}(\widetilde{\mathbf{x}}, y), \quad w(\widetilde{\mathbf{x}}) = p_y(\mathbf{x})^\alpha \big( \max\{p_y(\mathbf{x}) - p_y(\widetilde{\mathbf{x}}), 0\} \big)^\beta \quad (1)$$
where $p(\mathbf{x}) = \text{Softmax}\big(f_\theta(\mathbf{x})\big)$. Intuitively, the re-weighting factor $w(\widetilde{\mathbf{x}})$ emphasizes the samples of which (a) the original sample is confidently learned (*i.e.*, large $p_y(\mathbf{x})$) and (b) the gap of hardness between the original and the augmented sample is large (*i.e.,* large $p_y(\mathbf{x}) - p_y(\widetilde{\mathbf{x}})$).

**Rewarding not too different samples.** However, without proper regularization, the reward $\mathcal{R}_{\texttt{task}}$ may derive the augmentation policy to generate samples that are semantically meaningless, *e.g.*, out-of-distribution samples (Gong et al., 2020; Wei et al., 2020) or samples losing the characteristics of the target label $y$. See Figure 1(b) and 5(b) for a demonstration of such a behavior. Hence, for reducing such a potential risk, we further update the augmentation policy $\mathcal{T}_\phi$ to keep the semantic meaning of augmented sample $\widetilde{\mathbf{x}}$ be similar to the original $\mathbf{x}$. To this end, we first extract the semantics of sentence $\mathbf{x}$ from its contextualized embedding $\mathbf{o} = [o_1, \ldots, o_L]$ where $o_\ell$ is $\ell$-th output of the transformer model (Reimers et al., 2019; Zhang et al., 2020a). Then, a linear classifier $g_W$ is applied to classify whether the given embedding pair $(\mathbf{o}_1, \mathbf{o}_2)$ comes from the same sample ($y_{\texttt{pos}}$) or not ($y_{\texttt{neg}}$), respectively. With this classifier, we formulate the reward, $\mathcal{R}_{\texttt{sim}}$, for updating policy to correctly classify the given $(\mathbf{x}, \widetilde{\mathbf{x}})$ as positive, hence maintain the semantic of original $\mathbf{x}$ as follow:
$$\mathcal{R}_{\texttt{sim}}(\mathbf{x}, \widetilde{\mathbf{x}}) = -\ell_{\texttt{xe}}\big(p_W(\mathbf{m}, \widetilde{\mathbf{m}}), y_{\texttt{pos}}\big), \quad (2)$$
where $\ell_{\texttt{xe}}$ is the cross-entropy loss with $p_W(\mathbf{m}, \widetilde{\mathbf{m}}) = \text{Softmax}\big(g_W(\mathbf{m}, \widetilde{\mathbf{m}})\big)$ and
$$g_W(\mathbf{m}_1, \mathbf{m}_2) = W \cdot \text{Concat}(\mathbf{m}_1, \mathbf{m}_2, |\mathbf{m}_1 - \mathbf{m}_2|), \quad \mathbf{m} = \text{Mean-Pool}(\mathbf{o}) = \frac{1}{L}\sum_{\ell=1}^{L} o_\ell. \quad (3)$$

To train this linear classifier $g_W$, we add the following loss for training the classifier:
$$\mathcal{L}_{\texttt{sim}}(\mathbf{x}, \tilde{\mathbf{x}}) = \ell_{\texttt{xe}}\big(p_W(\mathbf{m}, \widetilde{\mathbf{m}}), y_{\texttt{pos}}\big) + \ell_{\texttt{xe}}\big(p_W(\mathbf{m}, \mathbf{n}), y_{\texttt{neg}}\big) \quad (4)$$
where $\mathbf{n}$ is the mean-pooled contextualized embedding of the other training sample except $\mathbf{x}$ itself.[2]

Furthermore, since the quality of contextualized embedding $\mathbf{o}$ is critical for our algorithm, we additionally constrain each $o_\ell$ to preserve the semantics of input tokens $x_\ell$. To this end, we additionally train our classifier with a word reconstruction loss $\mathcal{L}_{\texttt{recon}}$:
$$\mathcal{L}_{\texttt{recon}}(\mathbf{x}) = \sum_{l=1}^{L} \ell_{\texttt{xe}}(p_{\mathbf{V}}(o_l), x_l) \quad (5)$$

---

[2]Effect of using MLP for similarity loss is discussed in Appendix F.

---

**Algorithm 1** Learning to augment difficult, but dot too different (DND)

---

**Input:** Classifier from a pre-trained language model $f_\theta$, linear classifier $g_W$, data augmentation policy $\mathcal{T}_\phi$, training data $\mathcal{D}$, model learning rate $\eta_\theta$, policy learning rate $\eta_\phi$, update frequency $T_p$

**for** each iteration t **do**

    Draw a mini-batch $\mathcal{B} = \{(\mathbf{x}_i, y_i)_{i=1}^B\}$ from $\mathcal{D}$

    Construct augmented samples from policy $\tilde{\mathcal{B}} = \{(\tilde{\mathbf{x}}_i, y_i)_{i=1}^B\}$ where $\tilde{\mathbf{x}}_i = \tau(\mathbf{x}_i), \; \tau \sim \mathcal{T}_\phi$

    Update model to minimize $\mathcal{L}_{\texttt{task}}(\mathbf{x}_i, y_i) + w(\tilde{\mathbf{x}}_i)\mathcal{L}_{\texttt{task}}(\tilde{\mathbf{x}}_i, y_i) + \mathcal{L}_{\texttt{sim}}(\mathbf{x}_i, \tilde{\mathbf{x}}_i) + \lambda_{\texttt{r}}\mathcal{L}_{\texttt{recon}}(\mathbf{x}_i)$

    **if** t % $T_p$ = 0 **then**

        Update policy to maximize $\mathcal{R}_{\texttt{task}}(\tilde{\mathbf{x}}_i, y_i) + \lambda_{\texttt{s}}\mathcal{R}_{\texttt{sim}}(\mathbf{x}_i, \tilde{\mathbf{x}}_i)$

    **end if**

**end for**

---

where $p_{\mathbf{V}}(o) = \text{Softmax}(o\mathbf{V}^\top)$ and $\mathbf{V}$ is the word embeddings matrix within the transformer model. $\mathcal{L}_{\texttt{recon}}$ effectively preserves the information within output embeddings learned through a pre-training phase; hence, it could provide better output embeddings. More discussions are given in Appendix F.

In overall, our training loss of the classifier is as follow with a fixed hyper-parameter $\lambda_{\texttt{r}}$:

$$\mathcal{L}(\mathbf{x}, y) = \mathcal{L}_{\texttt{task}}(\mathbf{x}, y) + w(\tilde{\mathbf{x}})\mathcal{L}_{\texttt{task}}(\tilde{\mathbf{x}}, y) + \mathcal{L}_{\texttt{sim}}(\mathbf{x}, \tilde{\mathbf{x}}) + \lambda_{\texttt{r}}\mathcal{L}_{\texttt{recon}}(\mathbf{x}). \tag{6}$$

### 3.3 CONTINUOUS RELAXATION FOR TRAINING THE AUGMENTATION POLICY

Here, we describe the details on how the augmentation policy is optimized to maximize the non-differentiable reward function defined in Section 3.2. We first note that $\mathcal{T}_\phi$ consists of $T$ consecutive operations $\mathcal{O}_1, \ldots, \mathcal{O}_T$ where the operations are used to sample the augmentation from a pool of $K$ augmentations $\tau_1, \ldots, \tau_K$. Here, each operation $\mathcal{O}_t$ is parameterized as $\mathcal{O}_t(\boldsymbol{p}_t, \boldsymbol{\mu}_t)$, where $\boldsymbol{p} = [p_1, \ldots, p_K], \; \sum_k p_k = 1$ is the probability to sample the augmentation from the pool, and $\boldsymbol{\mu} = [\mu_1, \ldots, \mu_K]$ is the magnitudes used when each augmentation $\tau_k$ is sampled, *i.e.*, $\tau_k(\cdot; \mu_k) \sim \mathcal{O}(\boldsymbol{p}, \boldsymbol{\mu})$ with a probability $p_k$. Hence, as the non-differentiabiliy from operations breaks the gradient flow from the classifier (*e.g.,* sampling the augmentation), it is non-trivial how to optimize the parameters of augmentation policy $\phi = \{\boldsymbol{p}_t, \boldsymbol{\mu}_t\}_{t=1}^T$.

To this end, we adopt a continuous relaxation for each operation following (Hataya et al., 2020; Li et al., 2020), hence makes it be differentiable with respect to two trainable parameters of policy: probability $\boldsymbol{p}$ and magnitude $\boldsymbol{\mu}$. Here, the key idea is using 1) Gumbel-Softmax (Jang et al., 2017) instead of Categorical distribution for sampling with $\boldsymbol{p}$, and 2) straight-through estimator for $\boldsymbol{\mu}$ (Bengio et al., 2013). Therefore, by using this continuous relaxation, the classifier and the augmentation policy would be alternately trained in a fully differentiable manner. More discussions about the used augmentation pool and details are in Section 4.1 and Appendix A.3.

## 4 EXPERIMENTS

In this section, we evaluate our algorithm using two important NLP tasks: (1) text classification and (2) entailment tasks. We first describe the experimental setups in Section 4.1. In Section 4.2, we present empirical evaluations on DND and other baseline augmentations under both classification and entailment tasks. Finally, in Section 4.3, we provide additional analysis on our algorithm regarding (a) ablation of each component, (b) transferability of learned policy, (c) behavior of the learned augmentation policy. In Appendix D and E, additional qualitative and quantitative results are presented, respectively. Also, more ablation studies are given in Appendix F and G.

### 4.1 EXPERIMENTAL SETUPS

**Datasets and tasks.** For the text classification task, we use the following benchmark datasets: (1) News20 (Lang, 1995), (2) Review50 (Chen & Liu, 2014), and (3) CLINC150 (Larson et al., 2019) for topic classification, (4) IMDB (Maas et al., 2011) and SST-5 (Socher et al., 2013) for sentiment classification, and (6) TREC (Li & Roth, 2002) for question type classification. For datasets without given validation data, we use 10% of its training samples for the validation. With these datasets, we

Table 1: Test accuracy of RoBERTa classifiers after fine-tuning with each data augmentation method on 6 different text classification datasets under (5-shot / full) setups. All the values and error bars are mean and standard deviation across 3 random seeds. The best and the second best results are indicated in bold and underline, respectively.

| Method | IMDB | SST-5 | TREC | News20 | Review50 | CLINC150 |
|---|---|---|---|---|---|---|
| Vanilla | $59.1_{\pm3.2}$ / $95.2_{\pm0.1}$ | $25.3_{\pm2.5}$ / $57.1_{\pm0.5}$ | $69.3_{\pm6.3}$ / $97.1_{\pm0.4}$ | $46.3_{\pm2.1}$ / $82.0_{\pm0.4}$ | $45.1_{\pm0.5}$ / $71.4_{\pm0.1}$ | $86.4_{\pm0.3}$ / $95.5_{\pm0.2}$ |
| EDA | $56.7_{\pm2.1}$ / $95.3_{\pm0.1}$ | $27.4_{\pm1.8}$ / $57.2_{\pm0.5}$ | $71.8_{\pm3.5}$ / $97.0_{\pm0.2}$ | $47.8_{\pm2.4}$ / $83.3_{\pm0.6}$ | $46.4_{\pm1.1}$ / $72.4_{\pm0.1}$ | $87.1_{\pm0.8}$ / $96.1_{\pm0.1}$ |
| Adversarial | $\underline{66.1}_{\pm2.7}$ / $95.5_{\pm0.1}$ | $27.0_{\pm3.5}$ / $56.2_{\pm1.0}$ | $72.5_{\pm3.7}$ / $97.3_{\pm0.3}$ | $47.3_{\pm0.6}$ / $\underline{83.4}_{\pm0.2}$ | $45.7_{\pm0.6}$ / $73.0_{\pm0.2}$ | $87.4_{\pm0.3}$ / $96.0_{\pm0.2}$ |
| BERT-aug | $62.2_{\pm2.2}$ / $95.3_{\pm0.1}$ | $27.0_{\pm3.2}$ / $56.4_{\pm1.4}$ | $71.2_{\pm2.1}$ / $97.2_{\pm0.4}$ | $47.3_{\pm0.3}$ / $83.3_{\pm0.3}$ | $46.0_{\pm1.0}$ / $72.5_{\pm0.1}$ | $85.0_{\pm0.3}$ / $95.9_{\pm0.4}$ |
| Cutoff | $61.2_{\pm3.7}$ / $95.4_{\pm0.2}$ | $26.8_{\pm1.4}$ / $56.2_{\pm0.8}$ | $71.1_{\pm4.3}$ / $\underline{97.5}_{\pm0.5}$ | $\underline{48.9}_{\pm2.5}$ / $\underline{83.4}_{\pm0.4}$ | $47.2_{\pm2.0}$ / $72.7_{\pm0.2}$ | $86.7_{\pm0.7}$ / $96.0_{\pm0.1}$ |
| R3F | $61.3_{\pm2.4}$ / $95.5_{\pm0.1}$ | $25.3_{\pm3.5}$ / $57.1_{\pm1.1}$ | $70.5_{\pm4.3}$ / $\underline{97.5}_{\pm0.5}$ | $48.4_{\pm1.1}$ / $83.3_{\pm0.3}$ | $45.6_{\pm1.2}$ / $72.8_{\pm0.2}$ | $86.8_{\pm0.1}$ / $\underline{95.7}_{\pm0.1}$ |
| MixUp | $52.9_{\pm3.3}$ / $95.4_{\pm0.2}$ | $\underline{28.2}_{\pm2.2}$ / $57.0_{\pm1.2}$ | $71.6_{\pm1.0}$ / $97.1_{\pm0.3}$ | $48.5_{\pm0.4}$ / $82.9_{\pm0.1}$ | $\underline{47.9}_{\pm1.1}$ / $72.4_{\pm0.1}$ | $87.1_{\pm0.3}$ / $\underline{96.2}_{\pm0.3}$ |
| Back-trans | $59.6_{\pm3.8}$ / $95.3_{\pm0.1}$ | $25.7_{\pm2.7}$ / $\underline{57.5}_{\pm0.3}$ | $72.1_{\pm3.7}$ / $97.0_{\pm0.2}$ | $48.8_{\pm2.1}$ / $82.6_{\pm0.6}$ | $46.2_{\pm0.6}$ / $72.0_{\pm0.2}$ | $86.8_{\pm0.4}$ / $95.9_{\pm0.3}$ |
| Back&Adv | $65.2_{\pm2.3}$ / $\underline{95.6}_{\pm0.1}$ | $27.5_{\pm1.2}$ / $57.2_{\pm0.4}$ | $\underline{72.7}_{\pm4.2}$ / $97.1_{\pm0.3}$ | $47.7_{\pm2.2}$ / $83.2_{\pm0.5}$ | $45.5_{\pm1.0}$ / $\underline{73.3}_{\pm0.1}$ | $\underline{87.8}_{\pm0.1}$ / $\underline{96.2}_{\pm0.2}$ |
| Ours | $\mathbf{69.6}_{\pm2.7}$ / $\mathbf{95.7}_{\pm0.1}$ | $\mathbf{29.7}_{\pm2.0}$ / $\mathbf{58.3}_{\pm0.4}$ | $\mathbf{74.1}_{\pm1.4}$ / $\mathbf{98.0}_{\pm0.2}$ | $\mathbf{50.4}_{\pm1.4}$ / $\mathbf{85.2}_{\pm0.4}$ | $\mathbf{49.8}_{\pm0.5}$ / $\mathbf{74.9}_{\pm0.2}$ | $\mathbf{88.4}_{\pm0.4}$ / $\mathbf{96.6}_{\pm0.1}$ |

construct the 5-shot and class-imbalanced variants to further verify the effectiveness of augmentation for more challenging scenarios. We also validate the proposed method on GLUE benchmark (Wang et al., 2019), which is a collection of diverse natural language understanding tasks: 1 regression task (STS-B) and seven different classification tasks (RTE, MRPC, CoLA, SST-2, QNLI, QQP, and MNLI). As different metrics are used for each task in GLUE benchmark, we specify the used metric under the name of each task in Table 3. For the evaluation, the given development set is used. Detailed descriptions of tasks and construction of 5-shot and class-imbalanced datasets are in Appendix A.1.

**Baselines.** We compare our method with various baseline augmentations in NLP tasks. We first consider a naïve fine-tuning without data augmentation, denoted by (*a*) Vanilla, followed by various augmentation schemes; (*b*) EDA (Wei & Zou, 2019): synonym replacement, random insertion, random swap, and random deletion are randomly applied; (*c*) Adversarial (Jiang et al., 2020; Zhu et al., 2020): adversarial example is constructed by adding adversarial perturbation on the word embedding; (*d*) BERT-aug (Yi et al., 2021): input sentence is randomly masked, then filled with pre-trained BERT to construct new sample; (*e*) Cutoff (Shen et al., 2020): continuous span of word embedding is masked out similar to Dropout (Srivastava et al., 2014); (*f*) R3F (Aghajanyan et al., 2021): random noise are added to word embedding, then consistency loss is applied between original and perturbed sample; (*g*) MixUp (Guo et al., 2019): two independent samples are mixed on the word embedding and label space; (*h*) Back-trans (Xie et al., 2020): two translation models perform source to target and target to source translations; (*i*) Back&Adv (Qu et al., 2021): composition of back-translation and adversarial perturbation, that was identified as the best way for using multiple augmentations (Qu et al., 2021). Details on the implementations are presented in Appendix A.2.

**Training details.** All the experiments are conducted by fine-tuning RoBERTa-base (Liu et al., 2019) using Adam optimizer (Kingma & Ba, 2015) with a fixed learning rate 1e-5 and the default hyperparameters of Adam. For the text classification tasks, the model is fine-tuned using the specified augmentation method with batch size 8 for 15 epochs. For GLUE benchmark task, we commonly use batch size 16, except RTE task with batch size 8 following (Aghajanyan et al., 2021). Under each augmentation, the model is fine-tuned for 20 epochs on small tasks (RTE, MRPC, STS-B, CoLA) or 10 epochs on larger tasks (SST-2, QNLI, QQP, MNLI).

In the case of DND, we optimize the augmentation policy using Adam optimizer. We choose hyper-parameters in our method from a fixed set of candidates; the update frequency $T_p \in \{1, 5, 10, 100\}$, $\lambda_{\mathbf{s}} \in \{0.1, 0.5, 1.0, 2.0\}$, and $(\alpha, \beta) \in \{(0, 0), (0.5, 0.5)\}$ based on the validation set. Intuitively, the smaller datasets are beneficial for smaller $T_p$ and $\lambda_{\mathbf{s}}$. Other hyperparameters are fixed as $\lambda_{\mathbf{r}} = 0.05$ and policy's learning rate $\gamma_\phi$ as 1e-3. We use a simple two-layer MLPs for $W$. As the data augmentation pool, we include all the baseline augmentation except MixUp because it synthesizes a completely new sample by mixing two original samples. Hence, it is not clear whether it is meaningful to keep the similarity between original and augmented samples, which is the key component of DND. Also, to reduce the computational burden from the external models, we only update the probability to be sampled for the word-level augmentations (BERT-aug, Back-trans, and EDA) by generating augmented sentences before training following prior works (Wei & Zou, 2019; Yi et al., 2021; Qu et al., 2021). With this augmentation pool, our augmentation policy is composed of $T = 2$ consecutive operations. Also, to construct more diverse augmentations, we simultaneously train four

Table 2: Comparison of classification performance (bACC/GM) with RoBERTa classifiers on 3 different text classification datasets under 2 different class-imbalance ratio $\gamma_{\texttt{imb}}$. Except the gray-colored row, the classifiers are fine-tuned with over-sampled datasets augmented by each data augmentation method. All the values and error bars are mean and standard deviation across 3 random seeds. The best and the second best results are indicated in bold and underline, respectively.

| Method | SST-2 $\gamma_{\texttt{imb}} = 100$ | SST-2 $\gamma_{\texttt{imb}} = 20$ | News20 $\gamma_{\texttt{imb}} = 100$ | News20 $\gamma_{\texttt{imb}} = 20$ | Review50 $\gamma_{\texttt{imb}} = 100$ | Review50 $\gamma_{\texttt{imb}} = 20$ |
|---|---|---|---|---|---|---|
| Vanilla | $66.1_{\pm4.9}$ / $59.7_{\pm8.9}$ | $87.8_{\pm0.4}$ / $87.6_{\pm0.4}$ | $65.9_{\pm0.6}$ / $59.7_{\pm2.6}$ | $72.5_{\pm0.7}$ / $71.0_{\pm0.9}$ | $58.2_{\pm0.2}$ / $45.0_{\pm1.0}$ | $64.2_{\pm0.5}$ / $57.9_{\pm1.0}$ |
| Re-sample | $71.6_{\pm4.3}$ / $65.8_{\pm6.5}$ | $87.9_{\pm0.2}$ / $87.6_{\pm0.2}$ | $66.7_{\pm0.7}$ / $61.6_{\pm1.3}$ | $73.0_{\pm0.4}$ / $71.4_{\pm0.3}$ | $59.6_{\pm0.9}$ / $49.3_{\pm4.2}$ | $65.0_{\pm0.4}$ / $60.5_{\pm0.9}$ |
| EDA | $71.9_{\pm8.7}$ / $64.3_{\pm9.9}$ | $88.6_{\pm0.3}$ / $88.4_{\pm0.3}$ | $66.4_{\pm0.4}$ / $60.8_{\pm0.9}$ | $74.0_{\pm0.6}$ / $72.4_{\pm0.8}$ | $60.8_{\pm0.4}$ / $49.1_{\pm1.3}$ | $65.2_{\pm0.4}$ / $60.7_{\pm1.4}$ |
| Adversarial | $78.2_{\pm3.7}$ / $75.3_{\pm5.0}$ | $90.1_{\pm0.2}$ / $89.9_{\pm0.2}$ | $66.8_{\pm1.1}$ / $60.4_{\pm3.7}$ | $\underline{75.0}_{\pm0.7}$ / $\underline{73.9}_{\pm1.0}$ | $61.8_{\pm0.4}$ / $52.0_{\pm1.4}$ | $\underline{67.1}_{\pm0.2}$ / $\underline{63.4}_{\pm0.9}$ |
| BERT-aug | $67.9_{\pm6.2}$ / $60.1_{\pm6.3}$ | $89.8_{\pm0.2}$ / $89.7_{\pm0.4}$ | $66.5_{\pm0.3}$ / $59.8_{\pm0.2}$ | $74.1_{\pm0.7}$ / $72.7_{\pm0.7}$ | $60.7_{\pm0.6}$ / $52.6_{\pm0.6}$ | $65.3_{\pm0.5}$ / $60.8_{\pm0.9}$ |
| Cutoff | $77.2_{\pm3.8}$ / $74.2_{\pm5.5}$ | $89.4_{\pm0.3}$ / $89.3_{\pm0.3}$ | $67.0_{\pm0.6}$ / $61.4_{\pm1.0}$ | $73.7_{\pm1.1}$ / $71.9_{\pm1.0}$ | $60.8_{\pm0.4}$ / $50.2_{\pm0.7}$ | $65.6_{\pm0.6}$ / $61.1_{\pm1.4}$ |
| R3F | $72.5_{\pm1.3}$ / $67.9_{\pm2.4}$ | $89.1_{\pm1.6}$ / $88.9_{\pm1.8}$ | $66.5_{\pm0.8}$ / $60.9_{\pm2.1}$ | $73.9_{\pm0.4}$ / $72.4_{\pm0.4}$ | $59.6_{\pm0.6}$ / $47.2_{\pm2.7}$ | $65.2_{\pm0.5}$ / $60.5_{\pm1.1}$ |
| MixUp | $71.1_{\pm3.0}$ / $65.5_{\pm4.7}$ | $90.2_{\pm0.2}$ / $90.1_{\pm0.2}$ | $\underline{68.0}_{\pm0.2}$ / $\underline{64.1}_{\pm0.9}$ | $73.7_{\pm0.2}$ / $72.6_{\pm0.2}$ | $61.7_{\pm0.1}$ / $52.3_{\pm0.1}$ | $65.9_{\pm0.2}$ / $62.0_{\pm0.9}$ |
| Back-trans | $71.3_{\pm4.1}$ / $65.5_{\pm6.7}$ | $89.4_{\pm0.5}$ / $89.3_{\pm0.6}$ | $66.7_{\pm0.7}$ / $61.9_{\pm1.0}$ | $73.5_{\pm1.3}$ / $71.0_{\pm2.7}$ | $60.7_{\pm0.8}$ / $50.4_{\pm3.2}$ | $65.7_{\pm0.3}$ / $61.9_{\pm0.9}$ |
| Back&Adv | $\underline{78.8}_{\pm1.5}$ / $\underline{77.5}_{\pm3.0}$ | $\underline{90.3}_{\pm0.2}$ / $\underline{90.2}_{\pm0.2}$ | $67.0_{\pm0.5}$ / $60.7_{\pm0.6}$ | $74.0_{\pm0.3}$ / $71.7_{\pm1.5}$ | $\underline{61.9}_{\pm0.1}$ / $\underline{52.9}_{\pm0.2}$ | $66.7_{\pm0.2}$ / $63.0_{\pm0.5}$ |
| Ours | $\mathbf{84.3}_{\pm2.8}$ / $\mathbf{83.5}_{\pm3.5}$ | $\mathbf{91.1}_{\pm0.5}$ / $\mathbf{91.0}_{\pm0.6}$ | $\mathbf{69.8}_{\pm1.0}$ / $\mathbf{66.4}_{\pm1.0}$ | $\mathbf{76.8}_{\pm0.5}$ / $\mathbf{76.0}_{\pm0.7}$ | $\mathbf{63.9}_{\pm0.9}$ / $\mathbf{54.0}_{\pm1.7}$ | $\mathbf{68.4}_{\pm0.2}$ / $\mathbf{64.0}_{\pm0.8}$ |

Table 3: Evaluation results of vanilla (RoBERTa) classifiers with the GLUE development set after fine-tuning with each data augmentation method on the given task. All the values are mean across 3 random seeds and the standard deviation is presented in Appendix C. The best and the second best results are indicated in bold and underline, respectively. Gray-colored scores are those lower than the corresponding vanilla scores.

| Method (Metrics) | RTE (Acc) | MRPC (Acc) | STS-B (P/S) | CoLA (Mcc) | SST-2 (Acc) | QNLI (Acc) | QQP (Acc/F1) | MNLI-m/mm (Acc) |
|---|---|---|---|---|---|---|---|---|
| Vanilla | 79.4 | 89.62 | 90.72/90.42 | 62.2 | 94.46 | 92.64 | 90.87/88.17 | 87.34/86.85 |
| EDA | 77.8 | 90.03 | 90.14/89.92 | 55.8 | 94.96 | 92.56 | 91.41/88.50 | 87.46/87.31 |
| Adversarial | 80.5 | 91.00 | **91.08/90.68** | 63.5 | 94.88 | 93.18 | 91.50/88.58 | 87.91/87.84 |
| BERT-aug | 79.6 | 90.11 | 89.47/89.35 | 61.2 | 94.47 | 92.95 | 91.07/88.27 | 87.73/87.25 |
| Cutoff | 78.0 | 91.01 | 90.53/90.27 | 64.0 | 94.42 | 92.84 | 91.13/88.27 | 87.22/87.22 |
| R3F | 79.7 | 90.69 | 90.77/90.56 | 64.6 | 94.95 | 93.06 | 91.30/88.54 | 87.97/87.66 |
| MixUp | 79.8 | 90.08 | 90.46/90.16 | 64.3 | 94.50 | 92.69 | 90.98/88.06 | 87.50/87.02 |
| Back-trans | 79.9 | 90.20 | 90.54/90.33 | 58.8 | 94.53 | 92.64 | 90.82/88.05 | 87.50/87.06 |
| Back&Adv | 80.9 | 90.69 | 90.61/90.33 | 59.5 | 95.01 | 93.07 | 91.29/88.49 | 87.73/87.53 |
| Ours | **81.5** | **91.18** | 90.78/90.49 | **65.0** | **95.30** | **93.48** | **91.85/89.00** | **88.21/88.12** |

augmentation policies by randomly selecting one of them for each mini-batch. More details can be found in Appendix A.3.

## 4.2 EXPERIMENTAL RESULTS ON TEXT CLASSIFICATION AND GLUE BENCHMARK

To verify the effectiveness of DND, we compare DND with various data augmentation schemes in the scenario of fine-tuning RoBERTa-base model for each downstream task. Table 1 summarizes the experimental results on six text classification tasks, along with the challenging low-resource scenarios (*i.e.,* 5-shot). On the original full datasets, DND consistently outperforms the baseline augmentations for all datasets. To be specific, DND exhibits 16.45% relative test error reduction compared to the vanilla method in the average. Furthermore, compared to the previous best augmentation method for each dataset, DND exhibits 8.59% relative test error reduction in the average. In the case of challenging 5-shot datasets, our method again outperforms the strong baselines with a larger margin. These results show that our method successfully provides 'informative' learning signals to the training classifier regardless of the number of given labeled datasets.

Next, we evaluate DND on class-imbalanced datasets; more practical setups suffer from the limited training data. Here, we consider one more natural baseline, 'Re-sampling', that equally samples each class in a mini-batch. Then, it is combined with each augmentation by applying augmentation instead of using duplicated samples for re-sampling (for details, see Appendix A.1). For the evaluation, we report two popular metrics for class-imbalance: the arithmetic mean (bACC) and geometric mean (GM) over class-wise accuracy. In Table 2, one can observe that the gain from data augmentation is more significant compared to the case of balanced datasets, as the lack of diversity in minority classes would be addressed by it. However, our learning-based augmentation scheme further enlarges this

Table 4: Ablation study on different criteria for updating policy. Test accuracy of vanilla (RoBERTa) classifiers after fine-tuning across various augmentations and learning objectives on News20 and Review50 are compared. All the values and error bars are mean and standard deviation across 3 random seeds. The best and the second best results are indicated in bold and underline, respectively.

| Dataset | Vanilla | Fixed | Random | Difficult | Not Different | Validation | DND (no $w$) | DND |
|---|---|---|---|---|---|---|---|---|
| News20 | $82.49_{\pm0.23}$ | $84.06_{\pm0.19}$ | $84.58_{\pm0.16}$ | $83.96_{\pm0.12}$ | $84.65_{\pm0.13}$ | $\underline{85.08}_{\pm0.07}$ | $85.02_{\pm0.20}$ | $\mathbf{85.19}_{\pm0.35}$ |
| Review50 | $71.58_{\pm0.18}$ | $73.57_{\pm0.09}$ | $73.04_{\pm0.07}$ | $73.52_{\pm0.06}$ | $72.64_{\pm0.02}$ | $\underline{73.72}_{\pm0.22}$ | $74.13_{\pm0.16}$ | $\mathbf{74.87}_{\pm0.17}$ |

Table 5: Transferability of DND: test accuracy of the specified language model after fine-tuning across various augmentations. The augmentation policy learned with RoBERTa on News20 dataset is used as a base policy being transferred. All the values and error bars are mean and standard deviation across 3 random seeds. The best is indicated in bold.

| Model transfer | Dataset transfer | Vanilla | Fixed | Random | Transfer |
|---|---|---|---|---|---|
| RoBERTa | News20 → Review50 | $71.44_{\pm0.07}$ | $73.34_{\pm0.11}$ | $72.97_{\pm0.11}$ | $\mathbf{73.93}_{\pm0.01}$ |
| RoBERTa | News20 → Clinc150 | $95.48_{\pm0.21}$ | $96.50_{\pm0.07}$ | $96.31_{\pm0.04}$ | $\mathbf{96.63}_{\pm0.07}$ |
| RoBERTa → BERT | News20 | $84.11_{\pm0.58}$ | $84.65_{\pm0.12}$ | $86.01_{\pm0.06}$ | $\mathbf{86.32}_{\pm0.02}$ |
| RoBERTa → ALBERT | News20 | $79.70_{\pm0.23}$ | $80.03_{\pm0.10}$ | $79.37_{\pm0.31}$ | $\mathbf{80.48}_{\pm0.08}$ |

gain by constructing the augmented samples specialized for improving minority classes, based on its task-wise adaptability. Specifically, DND exhibits 20.33% and 9.12% relative test error reductions in the average compared to Re-sampling and the previous best augmentation method, respectively. Additionally, we perform t-test on the results in Tables 1 and 2 to validate the statistical significance of DND compared to other baselines. Here, we observe that the average p-value is 0.050, which is generally known to indicate the statistical significance of the results.

Furthermore, we evaluate DND on the GLUE benchmark, composed of 8 different tasks. Table 3 summarizes the corresponding results, and one can observe that DND consistently improves the RoBERTa-base model for all tasks; furthermore, DND outperforms the strong baselines for most cases. We remark that the existing augmentations are sometimes even harmful, as they can't consider the characteristics of each task. For example, the goal of CoLA task is classifying whether the given sentence is grammatically valid or not. Hence, this task is much sensitive to linguistic change, and it results in the degradation with the word-level augmentation methods. In contrast, our method automatically reflects such inherent characteristics of each task.

## 4.3 ADDITIONAL ANALYSES

**Ablation study.** To verify the effectiveness of the proposed reward function, we compare the following augmentation methods: (1) *Vanilla*: no data augmentation; (2) *Fixed*: fixed augmentation is applied throughout the fine-tuning. We choose the baseline augmentation that shows the best performance in Table 1; (3) *Random*: augmentation is randomly selected among the augmentation pool at each training step; (4) *Difficult*: policy is updated to increase the training loss of the augmented sample like Zhang et al. (2020c); (5) *Not Different*: policy is updated to increase the similarity between augmented and original samples; (6) *Validation*: policy is updated to maximize the validation accuracy directly like Auto-augment (Cubuk et al., 2019). To this end, we adopt LDM (Hu et al., 2019), which adapts an off-the-shelf reward learning algorithm from RL for joint data manipulation learning and model training. As LDM originally uses a different augmentation pool, we adapt it using the same augmentation pool with DND. (7) *DND (no $w$)*: policy is updated to increase both training loss and semantic similarity without re-weighting scheme in Eq. 1. Here, we commonly use the extra losses (Eq. 4 and Eq. 5) at training for the fair comparison. Effect of these losses is shown in Appendix F.

From Table 4, one can verify that the effectiveness of *Difficult* is quite limited due to it easily diverges to construct too different examples as shown in Figure 1(b) and 5(b). Also, the augmented samples from *Not Different* do not much improve the performance as the learning signals from these samples would be duplicated with the original samples. However, when we properly combine these rewards, it successfully provides 'informative' learning signals to the classifier (*DND (no $w$)*), and it can be further improved by the proposed re-weighting component (*DND*). Remarkably, it is observable that DND clearly outperforms *Validation*. These results demonstrate that our reward design is more effective than maximizing the validation accuracy for learning the augmentations. Furthermore, we

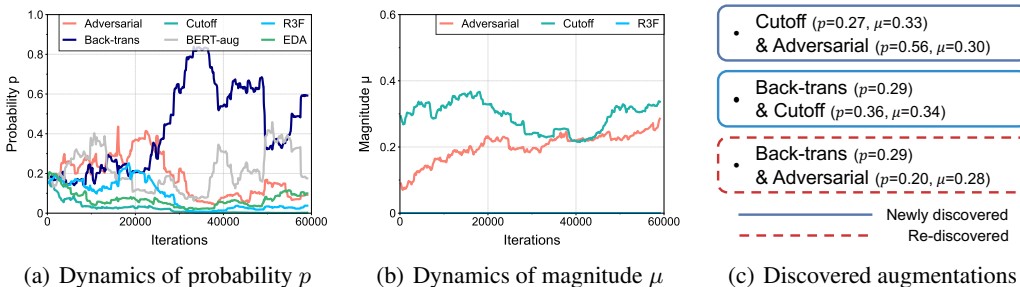

(a) Dynamics of probability $p$   (b) Dynamics of magnitude $\mu$   (c) Discovered augmentations

Figure 3: Analysis about the learned augmentation policy on Review50. (a, b) Dynamics of probability and magnitude during fine-tuning, respectively. (c) Top-3 representative augmentations, discovered by our augmentation policy. Best viewed in color.

emphasize that such methods maximizing validation accuracy usually require more computations than training loss-based methods due to an inherent bi-level optimization (Zhang et al., 2020c; Hataya et al., 2020); for example, LDM is 3x slower than DND under the same update frequency.

**Transferability across datasets and architectures.** Here, we provide additional experiments to verify the transferability of the learned policy with our method. To this end, we first save the status of the augmentation policy for each update during the training on the source data and then directly apply it to other tasks or models as dynamic augmentations. From Table 5, one can verify that the transferred augmentation outperforms the baseline augmentations and is even competitive with the performance of the jointly updated policy. Also, the learned policy could be transferred to the different models, such as BERT (Devlin et al., 2018) and ALBERT(Lan et al., 2019). These results imply that the proposed augmentation scheme would be greatly beneficial for successfully applying the data augmentation to the new task or model without the exhaustive search and update.

**More analysis on learned policy.** In Figure 3, we further present more analysis on the learned policy. Specifically, we focus on the parameters in the first operation $\mathcal{O}_1$. More detailed results can be found in Appendix B. We first note that the probability $p$ is dynamically changed at an early stage, which implies the exploration step is naturally occurred. After enough iterations, most of the augmentations are converged, and few candidates are only actively updated. In contrast, each augmentation's magnitude $\mu$ is progressively updated and almost converged at the last training iterations. This stability is actually the result of our reward function for similarity, as the policy quickly diverges without this (see Appendix B). Interestingly, from successful guidance of the proposed reward, we observe that our augmentation policy could re-discover the previously known best composition of augmentations (which was found by exhaustive grid search). Also, it can newly discover the task-specific augmentations as shown in Figure 3(c). It reveals the reason for success from DND, along with the efficiency of this method.

## 5   CONCLUSION

We propose DND, a simple and effective augmentation learning scheme for NLP tasks. We design of a novel reward function for optimizing augmentation policy to (a) construct the difficult augmented samples for providing 'informative' signals, while (b) preserving their semantic meaning to be not too different from the original for avoiding 'wrong' signals. From the experiments, we find our augmentation method to be much effective in challenging low-data and class-imbalanced regimes. Also, the learned augmentation policy transfers well across various tasks and models. As the augmentation policy learning for NLP tasks is under-explored in the literature, we expect our work to contribute to exploring this direction of research.

Furthermore, since the proposed concept of difficult but not too different augmentation is task- and domain-agnostic, we do believe that DND would benefit in other tasks (e.g., self-supervised learning) and domains (e.g., vision and graph) as well. On the other hand, incorporating the pre-trained models for augmentation learning in different domains (e.g., SimCLR (Chen et al., 2020)) would be an interesting future direction, as the fine-tuning of such pre-trained models have shown remarkable successes recently.

## ETHICS STATEMENT

Data augmentation is generally effective for improving the performance on the target task; hence it now becomes a de-facto standard for training DNNs, in various input domains (Cubuk et al., 2019; Park et al., 2019; You et al., 2020). However, it also has a potential risk to amplify the undesirable property of the model at the same time, *e.g.,* gender bias (Bordia & Bowman, 2019), since the enlarged diversity by data augmentation is proportional to the size of used samples for augmentation again. Since the practice of data augmentation is usually based on pre-defined fixed augmentations, it is hard to remove such inherency of undesirable properties explicitly.

In this respect, the learning-based augmentation scheme would be greatly beneficial, as it can easily alleviate this risk by setting the proper objective functions with respect to the desired property (Ziegler et al., 2019). For example, one could construct the 'fair' data augmentation scheme by jointly adding the reward for fairness and task accuracy, as we have explored in this work. Hence, we believe our proposed learning-based augmentation scheme would be a step towards improving a such fairness aspect in data augmentation, especially for NLP task.

## REPRODUCIBILITY STATEMENT

We describe the implementation details of the method in Section 4.1 and Appendix A.3. Also, we provide the details of the datasets and baselines in Appendix A.1 and A.2, respectively. We also provide our code in the supplementary material. All the used packages are along with the code. In our experiments, we use a single GPU (NVIDIA TITAN Xp) and 8 CPU cores (Intel Xeon E5-2630 v4).

## ACKNOWLEDGMENTS

This work was mainly supported by Institute of Information & communications Technology Planning & Evaluation (IITP) grant funded by the Korea government (MSIT) (No.2021-0-02068, Artificial Intelligence Innovation Hub; No.2019-0-00075, Artificial Intelligence Graduate School Program (KAIST)).

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

# Supplementary Material

## What Makes Better Augmentation Strategies?
## Augment Difficult but Not too Different

## A  DETAILS ON EXPERIMENTAL SETUPS

### A.1  DATASET

**Text classification task.** We have used the following 6 text classification datasets in Section 4. Here, we set the maximum length of the input sentence as 128 except IMDB dataset with 512. In case of 5-shot setup, it is further limited to 256 following (Hu et al., 2019).

(1) News20 (Lang, 1995) is a collection of 18,846 newsgroup documents, partitioned (nearly) evenly across 20 different newsgroups. Originally, it is given with 11,314 training samples and 7,532 test samples. News20 dataset is officially available at `http://qwone.com/~jason/20Newsgroups/`.

(2) Review50 (Chen & Liu, 2014) consists of Amazon reviews of 50 classes of electrical products. Each class has 1,000 reviews and it is divided into 700 and 300 reviews for training and test samples, respectively. Review50 dataset is officially available at `https://github.com/zhouyonglong/Mining-Topics-in-Documents`.

(3) CLINC150 (Larson et al., 2019) is a intent classification (text classification) dataset with 150 in-domain intent classes. In our experiments, we use the small version of CLINC150 which is officially provided at `https://github.com/clinc/oos-eval`. Here, each of the 150 in-domain intent classes is given with 50 training, 20 validation, and 30 test samples.

(4) SST-5 (Socher et al., 2013) is a dataset for sentiment classification on movie reviews, which are annotated with five labels; very positive, positive, neutral, negative, or very negative. In the given SST-5 dataset, there are 8,544 training, 1,101 validation, and 2,210 test samples. We have used SST-5 dataset at `https://pytorch.org/text/0.8.1/datasets.html#sst`.

(5) IMDB (Maas et al., 2011) is a dataset for binary sentiment classification on movie reviews. It is composed of 25,000 labeled training samples and 25,000 test samples. We use IMDB dataset provided at `https://pytorch.org/text/0.8.1/datasets.html#imdb`.

(6) TREC (Li & Roth, 2002) is a dataset for classifying the six question types, whether the question is about person, location, numeric information, and etc. There are 5,452 training samples and 500 test samples in TREC dataset at `https://pytorch.org/text/0.8.1/datasets.html#trec`.

**GLUE benchmark.** GLUE (Wang et al., 2019) is composed of 8 different tasks for natural language understanding: MNLI, MRPC, STS-B, CoLA, SST-2, QNLI, QQP, and RTE. Here, we also set the maximum length of the input sentence as 128 for each task, except CoLA with 64. Here, we offer a brief summary for each task. More detailed description for each task can be checked at `https://gluebenchmark.com/tasks`.

(1) MNLI is a binary entailment task which is composed of 393k training and 20k development samples. Given a pair of sentences, the goal is to predict whether the sentence is an *entailment, contradiction,* or *neutral* with respect to another one.

(2) MRPC is a binary classification task for deciding whether the given sentence pair is semantically equivalent or not. It is composed of 3.7k training and 408 development samples.

(3) STS-B is a regression task that scores (1 to 5) the semantic similarity between two sentences given in a pair. It is composed of 7k training and 1.5k development samples.

(4) CoLA is a binary single sentence classification task, where the goal is to predict whether the given English sentence is linguistically valid or not. It is composed of 8.5k training samples and 1k development samples.

(5) SST-2 is a binary single sentence classification task about movie reviews with human labels of their sentiment (*positive* or *negative*). It is composed of 67k training and 872 development samples.

(6) QNLI is a binary classification task that decides whether the given (question, sentence) pair contains the correct answer or not. It is composed of 108k training and 5.7k development samples.

(7) QQP is a binary classification task where the goal is to determine if two questions in a given pair are semantically equivalent. It is composed of 364k training and 40k development samples.

(8) RTE is a binary entailment task similar to MNLI, composed of 2.5k training and 276 development samples.

**Construction of 5-shot and class-imbalanced datasets.** In Section 4, we evaluate our method on 5-shot (in Table 1) and class-imbalanced regimes (in Table 2) which are artificially constructed from the existing benchmark datasets for text classification task. In case of 5-shot datasets, we randomly select five samples per each class to construct the datasets.

In case of class-imbalanced datasets, we consider "synthetically long-tailed" variants of SST-2, News20, and Review50 datasets in order to evaluate our algorithm under various levels of imbalance. For constructing the class-imbalanced training dataset, we assume the ordered numbers of data in each class as $N_1 \geq \cdots \geq N_K$. We use a single parameter $\gamma_{\texttt{imb}} \geq 1$, called the *imbalance ratio*, to control the class-imbalance of the dataset: once $\gamma_{\texttt{imb}}$ and $N_1$ are given, we set $N_k = N_1 \cdot \gamma_{\texttt{imb}}^{-\frac{k-1}{K-1}}$ so that $N_1 = \gamma_{\texttt{imb}} \cdot N_K$ as done by Cui et al. (2019). Namely, larger $\gamma_{\texttt{imb}}$ indicates more imbalanced class distribution. We use $N_1 = 1000$ for SST-10 and $N_1 = 400$ for News20 and Review50, respectively. For $\gamma_{\texttt{imb}}$, we use the specified numbers in Table 2.

**Re-sampling with data augmentation**: Following (Kim et al., 2020), we combine each data augmentation with batch-wise re-sampling, to handle the class-imbalance problem. Namely, in order to simulate the generation of $N_1 - N_k$ samples for any $k = 1, 2, \cdots, K$, we perform the data augmentation with probability $\frac{N_1 - N_{y_i}}{N_1}$, for all $i$ in a given class-balanced mini-batch $\mathcal{B} = \{(x_i, y_i)\}$.

## A.2 BASELINES

**(1) EDA** (Wei & Zou, 2019): Our implementation of EDA is based on the official implementation by the author at `https://https://github.com/jasonwei20`. 4 operations in EDA (synonym replacement, random insertion, random swap, random deletion) is randomly applied for each word with a probability $p_{\texttt{eda}}$, which is selected among $\{0.15, 0.3, 0.45\}$.

**(2) Adversarial** (Zhu et al., 2020): We choose hyper-parameter $\delta$ for the norm constraint of adversarial perturbation among $\{0.1, 0.2, 0.3\}$ based on its performance on the validation dataset. We construct the adversarial example with a single step update under $\ell_2$ norm.

**(3) BERT-aug** (Yi et al., 2021): Each token is randomly masked with a probability $p_{\texttt{mask}}$ and the masked token is replaced by the most likely token based on the output of pre-trained BERT model (`https://github.com/makcedward/nlpaug`). $p_{\texttt{mask}}$ is tuned among $\{0.15, 0.3, 0.45\}$.

**(4) Cutoff** (Shen et al., 2020): We choose hyper-parameter $\alpha$ for cutoff ratio among $\{0.1, 0.2, 0.3\}$ based on its performance on the validation dataset. It is observable that the dataset with shorter length usually benefit from the smaller value, and larger $\alpha$ is better for dataset with longer length. We adopt the official code at `https://github.com/dinghanshen/Cutoff`.

**(5) R3F** (Aghajanyan et al., 2021): We sample the noise $\epsilon$ from the uniform distribution $\mathcal{U} = [0, 1e\text{-}5]$, then it is added to the word embedding to construct the augmented samples. Unlike other augmentation, the consistency loss is used for training. The official code is available at `https://github.com/pytorch/fairseq`.

**(6) MixUp** (Guo et al., 2019): We adopt wordMixUp that mix two different sentences on their word embeddings. We use the hyper-parameter $\alpha_{\texttt{mix}}$ of mixing policy among $\{0.5, 1.0, 2.0\}$, which decides the mixing ratio $\lambda_{\texttt{mix}}$ where $\lambda_{\texttt{mix}} \sim \text{Beta}(\alpha_{\texttt{mix}}, \alpha_{\texttt{mix}})$.

**(7) Back-trans** (Xie et al., 2020): We use the en-de and de-en single models trained on WMT19 and they are available at `https://github.com/pytorch/fairseq`. We use the default setups: beam search with beam size 5, and keeping only top-1 hypothesis.

**(8) Back&Adv** (Qu et al., 2021): After Back-translation is applied to the original sample, adversarial perturbation is further added to its word embedding. Here, we choose hyper-parameter $\delta$ for the norm constraint of adversarial perturbation among $\{0.1, 0.2, 0.3\}$ based on its performance on the validation dataset.

## A.3   DND

For learning data augmentation via proposed DND, we use the following data augmentation pool: Cutoff (Shen et al., 2020), Back-translation (Xie et al., 2020), BERT-aug (Yi et al., 2021), Adversarial (Zhu et al., 2020), EDA (Wei & Zou, 2019) and random noise (used in R3F (Aghajanyan et al., 2021)). Here, we exclude MixUp in the augmentation pool because it synthesizes a completely new sample by mixing two original samples. Hence, it is not clear whether it is meaningful to keep the similarity between original and augmented samples, which is the key component of DND. Except for MixUp, we think any other augmentations can be in our augmentation pool. Also, although both BERT-aug and EDA have their own magnitude, we do not update these magnitudes, as it incurs the computational burden from the additional inference of external model (e.g., BERT) for each training iteration. Instead, we adopt the common practice in text augmentations (Wei & Zou, 2019; Yi et al., 2021; Qu et al., 2021), which generate the augmented sentences using the pre-determined magnitude before training the model, similar to the case of Back-trans. Hence, we only optimize the probability parameters for those word-level augmentations (BERT-aug, Back-trans, and EDA).

With this augmentation pool, our data augmentation is composed with $T = 2$ consecutive augmentations by finding their optimal probability and magnitude $p$ and $\mu$, respectively. Since the word-level augmentations (BERT-aug, Back-trans, and EDA) can't be applied after the embedding-level augmentations are applied, we limit the search space of $\mathcal{O}_2$ to the embedding-level augmentations. Also, to construct more diverse augmentations, we simultaneously train 4 augmentation policies by randomly selecting the one of them for each mini-batch. Also, we empirically observed that the training of augmentation policy can be improved by introducing the auxiliary operation (Hataya et al., 2020; Li et al., 2020) that decides whether to apply the selected augmentation or not, after sampling the augmentation from $p$.[3] Hence, we adopt this operation for training augmentation policy, but it is not necessary for the augmentation learning. Finally, we remark that other training losses for augmented samples, such as consistency loss (Jiang et al., 2020; Xie et al., 2020) or contrastive loss (Qu et al., 2021), can be used for training the model ($\mathcal{L}_{\texttt{task}}$) and the augmentation policy ($\mathcal{R}_{\texttt{task}}$), although we have used the most simplest cross-entropy loss, *i.e.*, $\mathcal{L}_{\texttt{task}} = \ell_{\texttt{xe}}$, to focus on the effect of augmentation itself.

---

[3]This operation is originally modeled by Bernoulli distribution, but can be approximated with Relaxed Bernoulli distribution as done in (Hataya et al., 2020; Li et al., 2020)

## B  MORE ANALYSIS ON LEARNED AUGMENTATION POLICY

In this section, we further provide the dynamics of learned augmentation policies with respect to probability $p$ and magnitude $\mu$ under two reward functions: (1) *Difficult* , which was defined in Section 4.3, and (2) Difficult, but not too Different (Ours). As we have partially shown in Section 4.3, our reward function successfully guides the policy to learn the effective augmentation with progressive and steady updates on the parameters (in Figure 4). In contrast, as one can see in Figure 5, rewarding the augmentation policy just to maximize the training loss would mislead policy to generate samples that are semantically meaningless, *e.g.,* out-of-distribution samples or samples losing the characteristics of the target label $y$. Specifically, in Figure 5(b), one can actually see evidence for this; the size of adversarial perturbation becomes too large. Hence, this empirical observation again shows the importance of proper reward function to update augmentation policy and the validity of the proposed components in DND (maximizing semantic similarity and re-weighting scheme).

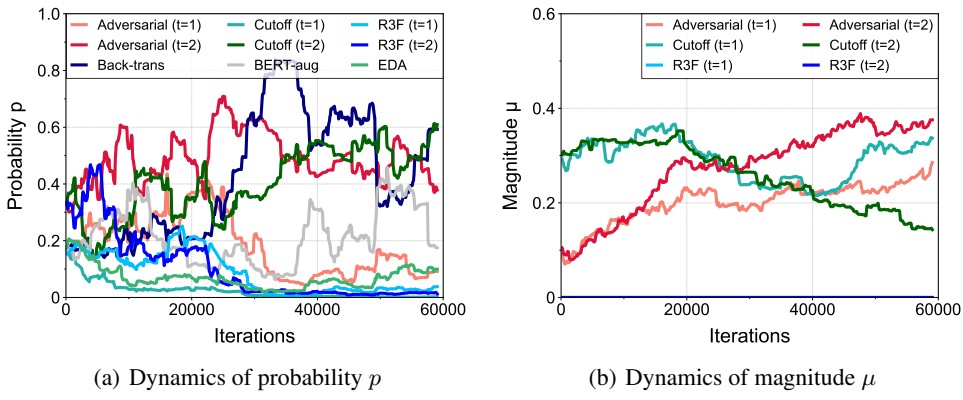

(a) Dynamics of probability $p$           (b) Dynamics of magnitude $\mu$

Figure 4: Dynamics of learned augmentation policy via DND.

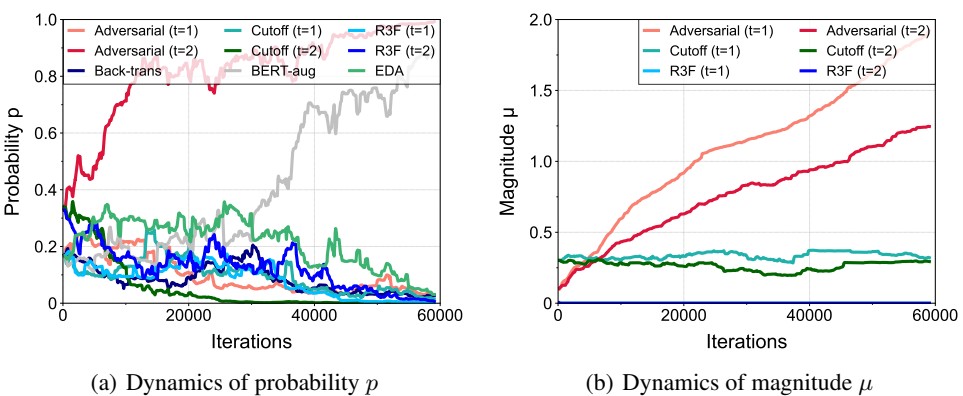

(a) Dynamics of probability $p$           (b) Dynamics of magnitude $\mu$

Figure 5: Dynamics of learned augmentation policy via *Difficult*.

## C  COMPLEMENTARY RESULTS ON GLUE BENCHMARK

In this section, we provide the complementary results of GLUE benchmark (Wang et al., 2019). First, Table 6 shows the results on the development set including the standard deviation across 3 random seeds, have omitted for better presentation of the experimental results. Next, we also validate the effectiveness of ours method by comparing it with vanilla on the test set, which is available at `https://gluebenchmark.com/submit`. As shown in Table 7, ours improves the performance of vanilla classifier in most cases.

Table 6: Evaluation results of vanilla (RoBERTa) classifiers with the GLUE development set after fine-tuning with each data augmentation method on the given task. All the values and error bars are mean and standard deviation across 3 random seeds. The best and the second best results are indicated in bold and underline, respectively.

| Method (Metrics) | RTE (Acc) | MRPC (Acc) | STS-B (P/S) | CoLA (Mcc) | SST-2 (Acc) | QNLI (Acc) | QQP (Acc/F1) | MNLI-m/mm (Acc) |
|---|---|---|---|---|---|---|---|---|
| Vanilla | 79.4 ±1.03 | 89.62 ±0.42 | 90.72/90.42 ±0.12/±0.15 | 62.2 ±1.27 | 94.46 ±0.14 | 92.64 ±0.07 | 90.87/88.17 ±0.03/±0.07 | 87.34/86.85 ±0.08/±0.06 |
| EDA | 77.8 ±0.74 | 90.03 ±0.12 | 90.14/89.92 ±0.13/±0.12 | 55.8 ±1.29 | 94.96 ±0.07 | 92.56 ±0.12 | 91.41/88.50 ±0.10/±0.20 | 87.46/87.31 ±0.01/±0.17 |
| Adversarial | 80.5 ±0.74 | 91.00 ±0.31 | **91.08/90.68** ±0.08/±0.04 | 63.5 ±0.62 | 94.88 ±0.21 | 93.18 ±0.11 | 91.50/88.58 ±0.01/±0.15 | 87.91/87.84 ±0.02/±0.14 |
| BERT-aug | 79.6 ±1.28 | 90.11 ±0.70 | 89.47/89.35 ±0.25/±0.33 | 61.2 ±1.25 | 94.27 ±0.28 | 92.95 ±0.13 | 91.07/88.27 ±0.03/±0.12 | 87.73/87.25 ±0.19/±0.02 |
| Cutoff | 78.0 ±0.76 | 91.01 ±0.51 | 90.53/90.27 ±0.29/±0.23 | 64.0 ±1.27 | 94.42 ±0.22 | 92.84 ±0.03 | 91.13/88.27 ±0.02/±0.08 | 87.22/87.22 ±0.05/±0.02 |
| R3F | 79.7 ±1.18 | 90.69 ±0.35 | 90.77/90.56 ±0.08/±0.06 | 64.6 ±0.45 | 94.95 ±0.06 | 93.06 ±0.01 | 91.30/88.54 ±0.01/±0.07 | 87.97/87.66 ±0.28/±0.27 |
| MixUp | 79.8 ±0.72 | 90.08 ±0.62 | 90.46/90.16 ±0.16/±0.16 | 64.3 ±0.83 | 94.50 ±0.28 | 92.69 ±0.14 | 90.98/88.06 ±0.05/±0.03 | 87.50/87.02 ±0.07/±0.05 |
| Back-trans | 79.9 ±0.90 | 90.20 ±0.20 | 90.54/90.33 ±0.12/±0.17 | 58.8 ±1.05 | 94.53 ±0.20 | 92.64 ±0.21 | 90.82/88.05 ±0.02/±0.01 | 87.50/87.06 ±0.10/±0.12 |
| Back&Adv | 80.9 ±1.03 | 90.69 ±0.53 | 90.61/90.33 ±0.07/±0.05 | 59.5 ±1.01 | 95.01 ±0.06 | 93.07 ±0.14 | 91.29/88.49 ±0.01/±0.03 | 87.73/87.53 ±0.13/±0.12 |
| Ours | **81.5** ±0.87 | **91.18** ±0.12 | 90.78/90.49 ±0.04/±0.02 | **65.0** ±0.25 | **95.30** ±0.12 | **93.48** ±0.12 | **91.85/89.00** ±0.05/±0.07 | **88.21/88.12** ±0.12/±0.08 |

Table 7: Evaluation results of vanilla (RoBERTa) classifiers with the GLUE test set after fine-tuning with each data augmentation method on the given task. For evaluation, we submit the prediction averaged across 3 random seeds. The better results are indicated in bold.

| Method (Metrics) | RTE (Acc) | MRPC (Acc) | STS-B (P/S) | CoLA (Mcc) | SST-2 (Acc) | QNLI (Acc) | QQP (Acc/F1) | MNLI-m/mm (Acc) |
|---|---|---|---|---|---|---|---|---|
| Vanilla | 72.4 | 87.1 | **88.9/88.2** | 58.8 | 94.8 | 92.3 | 89.6/71.8 | 86.9/86.4 |
| Ours | **74.0** | **87.7** | 88.6/87.8 | **60.1** | **95.3** | **93.3** | **89.7/72.8** | **87.9/87.2** |

# D    MORE QUALITATIVE RESULTS

**Difficult, but not different samples.** In this section, to further verify whether the augmented samples with the desired property are constructed from the learned policy, we construct the augmented samples using each augmentation method and compare the average difficulty and similarity in Figure 6(a). We observe that the augmented samples from our augmentation policy actually have a desired property: they show significantly low confidence, *i.e.*, more difficult, while it moderately preserves the semantic of original samples. Furthermore, we present the augmented samples from BERT-aug by varying the difficulty and similarity in Figure 6(b). Here, one can verify that the *difficulty* is solely not enough as too many changes can alter its semantic. Also, the *similarity* is solely not enough as it tends to construct too easy samples which can't provide 'informative' signals to the training model. However, by considering two aspects jointly, reasonable and effective augmented samples can be constructed. More qualitative results are presented in Figure 8. Also, the plot of augmented samples by each augmentation scheme is presented in Figure 7.

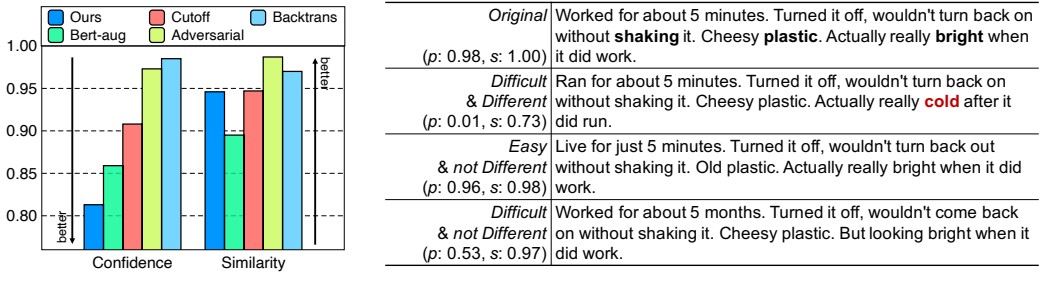

(a) *Difficulty* and *Similarity*                    (b) Example of augmented samples (Label: 'Lamp')

Figure 6: Qualitative results of difficult, but not too different samples; (a) Average *difficulty* (difficulty ↑ = confidence ↓) and *similarity* of the augmented samples under each augmentation method on News20. (b) Real examples on Review50. Different criteria are used for the selection, and two most salient words of original sentence are remarked in **bold**. Critically altered words are colored in red. Here, $p$ and $s$ represent the confidence and similarity, respectively. Best viewed in color.

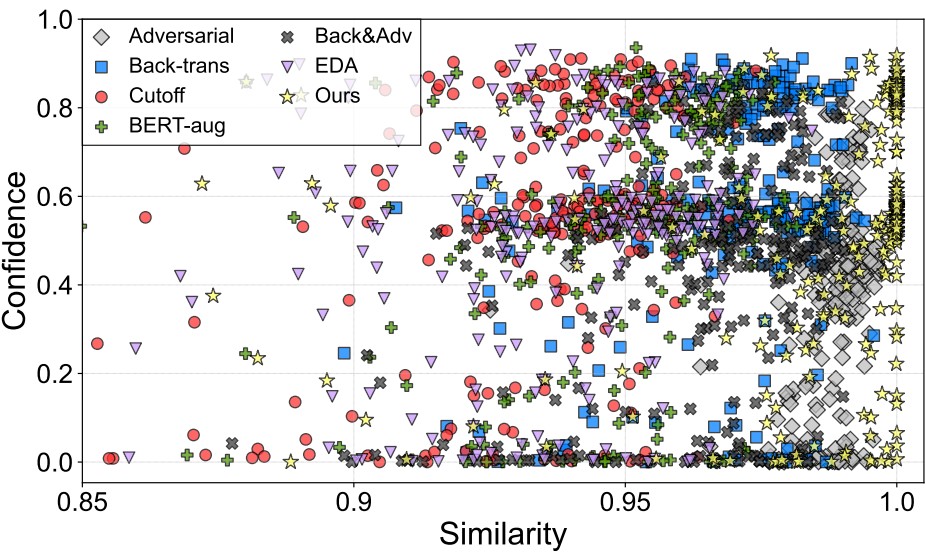

Figure 7: Difficulty (difficulty ↑ = confidence ↓) and similarity of the augmented samples under each augmentation method on Review50. For the better visualization, 250 training examples are randomly selected and then augmented.

| | |
|---|---|
| Original
(*p*: 0.99, *s*: 1.00) | Everything seems to work fine, the only **SDHC device** I have is my N82 and they work great together. N95 8**GB** eat your heart out! |
| *Difficult*
& *Different*
(*p*: 0.01, *s*: 0.20) | Everything seems to work fine, the closest SDHC **chips** I have is my N82 but they work great together. N95 **can** eat your brain out! |
| *Easy*
& *not Different*
(*p*: 0.99, *s*: 0.91) | Everything seems to work fine, the only SDHC **card** I have is an N82 where they work great together. You 8GB eat your heart alive! |
| *Difficult*
& *not Different*
(*p*: 0.54, *s*: 0.76) | Everything continues to be fine, the best SDHC device I have is my N82 and you work great together. N95 8GB eat your brain out! |

**Label: "Memory Card"**

| | |
|---|---|
| Original
(*p*: 0.97, *s*: 1.00) | I am using this card in my new build and it works flawlessly. I love **ASUS** products anyway, and this has not disappointed. I highly recommend this card if you are looking for a **wireless adapter**. |
| *Difficult*
& *Different*
(*p*: 0.00, *s*: 0.82) | I am using one card in my new build but it works flawlessly. I love ASUS products anyway, also this has not disappointed. I highly recommend this card when you are looking for a **play phone**. |
| *Easy*
& *not Different*
(*p*: 0.98, *s*: 0.96) | I am using this chip in a new build so it works flawlessly. I love ASUS products anyway, and this child not disappointed. I highly recommend a board if you are looking for a wireless adapter. |
| *Difficult*
& *not Different*
(*p*: 0.66, *s*: 0.96) | I am using this card in my new build and it operates flawlessly. I love ASUS as well, and this has not disappointed. Should highly recommend this card if you are pick up a wireless adapter. |

**Label: "Memory Card"**

| | |
|---|---|
| Original
(*p*: 0.99, *s*: 1.00) | When I bought this product I didn't read -LRB- and I'm pretty sure it wasn't clarified anywhere -RRB- that this **remote** wouldn't work in my **Nikon** D3100. So, I haven't been able to test it. |
| *Difficult*
& *Different*
(*p*: 0.01, *s*: 0.79) | When when bought my product I didn't read -LRB- and I'b pretty sure it **wouldn't looking** anywhere -RRB- that this **product** wouldn't work in my Nikon d3100. So, I shouldn't been able to test **myself**. |
| *Easy*
& *not Different*
(*p*: 0.99, *s*: 0.95) | When I sold this product I didn't read -but- and I'm pretty sure script wasn't clarified anywhere -RRB- that this remote wouldn't work against my Nikon d3100. So, I haven's got allowed to test it. |
| *Difficult*
& *not Different*
(*p*: 0.59, *s*: 0.96) | Until I bought this thing I didn't read -LRB- and I'm pretty sure it wasn't clarified anywhere -RRB- that phone remote wouldn't appear in my **new** d3100. So, I haven't been able to test it. |

**Label: "Remote Control"**

| | |
|---|---|
| Original
(*p*: 0.99, *s*: 1.00) | I have two of these **systems**, bought one for work and one for home. They work great. Yes, one of them had the same trouble as you may read in some comments. I found if you eject the **disks** before you shut it off, they never get stuck in it. the one machine where they did get stuck, I unplugged it for two days and plugged it back in and it spit out the **cd**. |
| *Difficult*
& *Different*
(*p*: 0.03, *s*: 0.90) | I have two of these **discs**, bought one for work and other for rent. Things were great. Yes, one of them had the biggest trouble with you may read in some records. But found if you eject the disks before you shut it off, they never get stuck in it. at one machine where they did get lost, I unplugged it for five days and plugged it back in and it spit out the **tape**. |
| *Easy*
& *not Different*
(*p*: 0.99, *s*: 0.96) | He released two of these systems, bought one for businesses and one for home. They work great. Yes, one of them had those same trouble as you may add in some comments. I found if you heard the disks before you got it on, they never get stuck in them. the one machine where they did were caught, I unplugged it for two days only plugged it back in and it blew out the cd. |
| *Difficult*
& *not Different*
(*p*: 0.69, *s*: 0.94) | I edited two of his systems, bought one for work and one from home. They work great. Yes, one of them had the same trouble like you may read or some comments. We found if you tried three disks before you shut it off, they all get stuck in it. if one machine where it did became stuck, I unplugged it for two days to plugged it back in and it spit out the cd. |

**Label: "Home Theater System"**

Figure 8: Examples of augmented samples on Review50 (Chen & Liu, 2014). Different criteria are used for the selection, and three most salient words of original sentence are remarked in **bold**. Critically altered words are colored in **red**. Here, *p* and *s* represent the confidence and similarity, respectively.

# E    MORE QUANTITATIVE RESULTS

In this section, we present more quantitative results about the proposed DND. Here, all experiments are conducted on News20 (Lang, 1995) dataset. Also, all the values and error bars are mean and standard deviation across 3 random seeds. The best result is indicated in bold.

Table 8: Experimental results to verify the compatibility among different language models. Test accuracy of specified classifiers after fine-tuning across various data augmentations are compared.

| Method | RoBERTa-large | BERT-base | ALBERT-base |
|---|---|---|---|
| Vanilla | $83.53_{\pm0.57}$ | $83.44_{\pm0.30}$ | $78.79_{\pm0.73}$ |
| Adversarial | $84.14_{\pm0.26}$ | $84.55_{\pm0.09}$ | $79.73_{\pm0.08}$ |
| Cutoff | $85.09_{\pm0.14}$ | $85.49_{\pm0.35}$ | $79.66_{\pm0.94}$ |
| R3F | $84.58_{\pm0.13}$ | $84.53_{\pm0.17}$ | $78.99_{\pm0.08}$ |
| Ours | $\mathbf{85.79}_{\pm0.25}$ | $\mathbf{85.93}_{\pm0.48}$ | $\mathbf{80.42}_{\pm0.36}$ |

**Compatibility of DND about model choice.** In Section 4, we mainly conduct experiments with RoBERTa-base(Liu et al., 2019) model. Hence, to verify the compatibility of DND about model choice, we further provide the experimental results under the different choice of pre-trained language models: RoBERTa-large (Liu et al., 2019), BERT-base (Devlin et al., 2018), and ALBERT-base (Lan et al., 2019). RoBERTa-large has 24 Transformer blocks with 1024 hidden size and 16 self-attention heads (total parameters=340M). BERT-base has the same architecture with RoBERTa-base (12 blocks with 768 hidden size and 12 self-attention heads, hence 110M total parameters), but it's pre-trained in a different way. ALBERT-base had an identical architecture with BERT-base except 128 hidden size (total parameters=12M). In overall, the model size has following order: RoBERTa-large > RoBERTa-base = BERT-base > ALBERT-base. In Table 8, we compare DND with the following 3 augmentations that are top-3 baselines in Table 1: Adversarial (Zhu et al., 2020), Cutoff (Shen et al., 2020), and R3F (Aghajanyan et al., 2021). Here, DND outperforms all baselines and it shows that DND successfully learns the effective data augmentation regardless of the model choice.

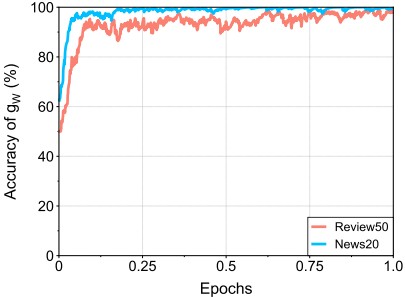 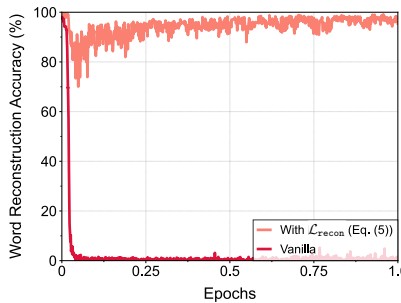

Figure 9: Accuracy of $g_W$ (Eq. 3) across training iterations on News50 and Review50 datasets. Here, we only report the results during one epoch for better visualization.

Figure 10: Accuracy of word reconstruction (Eq. 5) across training iterations on Review 50 dataset. Here, we only report the results during one epoch for better visualization.

## F    DETAILED ANALYSIS ON EXTRA LOSSES

In this section, we provide the detailed analysis on two extra losses $\mathcal{L}_{\texttt{sim}}$ (Eq. 4) and $\mathcal{L}_{\texttt{recon}}$ (Eq. 5) used when training the model, especially to investigate the following:

- For $\mathcal{L}_{\texttt{sim}}$ (and $\mathcal{R}_{\texttt{sim}}$), using $g_W$ with MLP is better than simpler choices such as cosine-similarity or $\ell_2$ distance?
- What is the contribution of $\mathcal{L}_{\texttt{recon}}$ in DND?

We remark that $\mathcal{L}_{\texttt{sim}}$ is introduced to train the linear classifier $g_W$, which is used for rewarding not too different samples with $\mathcal{R}_{\texttt{sim}}$ (Eq. 2). $\mathcal{L}_{\texttt{recon}}$ is introduced to preserve the original semantics of input token, as the quality of contextualized output embedding is critical for our algorithm. Here, all experiments are conducted on News20 (Lang, 1995) and Review50 (Chen & Liu, 2014) datasets. Also, all the values and error bars are mean and standard deviation across three random seeds. The best is indicated in bold.

**Discussions with simpler choices for similarity loss.** Although other simpler choices, such as cosine-similarity or $\ell_2$ distance on the output embeddings, are considerable as the similarity loss, using MLP allows a better trade-off between the original task performance and learning the similarity between the embeddings. Namely, the simpler choices could over-regularize the output embedding and result in an underfitting for the target task. However, by leveraging MLP, more information for the target task can be formed and maintained in the output embeddings. Hence, $g_W$ with MLP can get properly trained during the fine-tuning for the target task, even with a small number of training iterations. For example, on Review50 dataset, we empirically observed that $g_W$ achieves 98.1% accuracy only after one epoch of training, as shown in Figure 9. To empirically validate this intuition, we conduct the following experiments with DND using 1) cosine similarity and 2) $\ell_2$ distance between $\mathbf{m}_1$ and $\mathbf{m}_2$, instead of using MLP. As shown in the Table 9, the performance with the simpler choices actually decreases compared to using MLP. We also remark that the effectiveness of MLP for measuring the similarity on the embeddings has been demonstrated in the other context as well, e.g., SimCLR (Chen et al., 2020).

Table 9: Ablation study on different choices for similarity loss.

| Dataset | Cosine similarity | $\ell_2$ distance | MLP |
|---|---|---|---|
| News20 | $84.42_{\pm 0.10}$ | $83.99_{\pm 0.29}$ | $\mathbf{85.19}_{\pm 0.35}$ |
| Review50 | $74.04_{\pm 0.07}$ | $73.81_{\pm 0.17}$ | $\mathbf{74.87}_{\pm 0.17}$ |

**More intuition and effect of reconstruction loss.** First, we present a more detailed description how reconstruction loss works. Input token $x \in \{0, 1\}^V$ is a $V$-dimensional one-hot vector with a vocabulary size $V$. $\mathbf{V} \in \mathbb{R}^{V \times d}$ is the word embeddings matrix of pre-trained transformer model that maps $x$ to $d$-dimensional input embeddings $z \in \mathbb{R}^d$. This $z$ is feed-forwarded to the remaining

parts of the transformer (e.g., self-attention blocks) and the corresponding output embeddings is $o \in \mathbb{R}^d$. Hence, the resulting word reconstruction loss can be formulated as $\ell_{\mathtt{xe}}(p_{\mathbf{V}}(o), x)$ (Eq. 5) where $p_{\mathbf{V}}(o) = \mathrm{Softmax}(o\mathbf{V}^\top)$. Namely, it is predicting one-hot label (given by input token $x$) from the dot product of output embeddings $o$ and word embedding matrix $\mathbf{V}$.

Intuitively, reconstruction loss $\mathcal{L}_{\mathtt{recon}}$ effectively preserves the information within output embeddings learned through the pre-training phase; hence, it could provide a better output embedding. We empirically observed that the word reconstruction (i.e., prediction) from output embeddings becomes impossible as the model is fine-tuned in the target task (See Figure 10). We remark that this word reconstruction was almost perfectly possible before the fine-tuning, as the language model is pre-trained using a similar reconstruction loss (e.g., masked token prediction in BERT). Thus, this observation implies that this semantic information is lost. Our reconstruction loss effectively prevents such information loss and improves the quality of output embeddings. To empirically demonstrate the effectiveness of this term, we further provide the experimental results of DND without reconstruction loss (i.e., $\lambda_r = 0$). As shown in Table 10, the reconstruction loss clearly improves the performance of DND by providing better output embeddings.

Table 10: Ablation study on the effect of reconstruction loss

| Dataset | $\lambda_r = 0$ | $\lambda_r = 0.05$ |
|---|---|---|
| News20 | $84.88_{\pm 0.19}$ | $\mathbf{85.19}_{\pm 0.35}$ |
| Review50 | $74.52_{\pm 0.03}$ | $\mathbf{74.87}_{\pm 0.17}$ |

**Effect of extra losses.** While the extra losses $\mathcal{L}_{\mathtt{sim}}$ (Eq. 4) and $\mathcal{L}_{\mathtt{recon}}$ (Eq. 5) are introduced for updating policy with DND, these losses could contribute to the learning of better representations and lead to the improvement themselves. To verify this, we conduct the additional experiments about these auxiliary losses. Here, when each loss is applied, the fixed coefficients are used: 1 for $\mathcal{L}_{\mathtt{sim}}$ and 0.05 for $\mathcal{L}_{\mathtt{recon}}$. In Table 11, it is verified that each of the proposed losses could improve the performance contributing to the learning of better representations, and two auxiliary losses are complementary to each other.

Table 11: Effect of extra losses $\mathcal{L}_{\mathtt{sim}}$ and $\mathcal{L}_{\mathtt{recon}}$.

| Dataset | $\mathcal{L}_{\mathtt{sim}}$ | $\mathcal{L}_{\mathtt{recon}}$ | Cutoff | Back-translation | Adversarial |
|---|---|---|---|---|---|
| News20 | - | - | $83.36_{\pm 0.41}$ | $82.57_{\pm 0.60}$ | $83.37_{\pm 0.23}$ |
| | ✓ | - | $83.74_{\pm 0.42}$ | $82.77_{\pm 0.49}$ | $83.63_{\pm 0.15}$ |
| | - | ✓ | $83.52_{\pm 0.13}$ | $82.86_{\pm 0.47}$ | $83.44_{\pm 0.08}$ |
| | ✓ | ✓ | $\mathbf{84.06}_{\pm 0.19}$ | $\mathbf{83.01}_{\pm 0.23}$ | $\mathbf{83.79}_{\pm 0.56}$ |

## G    MORE ABLATION STUDY

In this section, we provide more ablation study on the design choices of DND. Here, all experiments are conducted on News20 (Lang, 1995) and Review50 (Chen & Liu, 2014) datasets. Also, the same values are used for the others except for the specified hyper-parameters. All the values and error bars are mean and standard deviation across three random seeds. The results with originally used values are underlined.

**Number of augmentation policies** $n_{\mathcal{T}}$. As we have mentioned in Appendix A.3, we simultaneously train multiple augmentation policies ($n_{\mathcal{T}} = 4$). To see the effect of this, we compare the cases with less ($n_{\mathcal{T}} = 1$) and more ($n_{\mathcal{T}} = 10$) number of policies in Table 12. Overall, the improvement from DND is not sensitive to the number of augmentation policies, but using multiple policies shows slightly better results as it would construct more diverse augmentations.

Table 12: Effect of different number of augmentation policies.

| Dataset | $n_{\mathcal{T}} = 1$ | $n_{\mathcal{T}} = 4$ | $n_{\mathcal{T}} = 10$ |
|---|---|---|---|
| News20 | $84.93_{\pm 0.05}$ | $\underline{85.19}_{\pm 0.35}$ | $85.16_{\pm 0.08}$ |
| Review50 | $74.84_{\pm 0.18}$ | $\underline{74.87}_{\pm 0.17}$ | $74.90_{\pm 0.17}$ |

**Number of operations** $T$. As described in Section 4.1, our augmentation policy is composed of two consecutive operations. Here, we compare this choice ($T = 2$) with less ($T = 1$) and more ($T = 3, 4$) number of operations. As shown in Table 13, one can verify that a single operation is not enough to construct complexly augmented samples; hence the empirical gain is relatively small compared to the original choice. With more operations, almost saturated improvement is observed. Remarkably, similar tendencies are also observed in vision tasks (Hataya et al., 2020; Cubuk et al., 2020).

Table 13: Effect of different number of operations $T$.

| Dataset | $T = 1$ | $T = 2$ | $T = 3$ | $T = 4$ |
|---|---|---|---|---|
| News20 | $84.68_{\pm 0.41}$ | $\underline{85.19}_{\pm 0.35}$ | $85.19_{\pm 0.25}$ | $85.22_{\pm 0.10}$ |
| Review50 | $74.24_{\pm 0.07}$ | $\underline{74.87}_{\pm 0.17}$ | $74.93_{\pm 0.07}$ | $75.01_{\pm 0.02}$ |

**Different values for** $\alpha$ **and** $\beta$. We additionally provide the results with different values for $\alpha$ and $\beta$, which are introduced for re-weighting scheme in $\mathcal{R}_{\texttt{task}}$ (Eq. 1). In Table 14, it is observable that each component of the re-weighting scheme is solely effective, and the gain from re-weighting is not much sensitive to the choice of $\alpha$ and $\beta$. We remark the original choice $(\alpha, \beta) = (0.5, 0.5)$ shows the best empirical gains.

Table 14: Effect of different values for hyper-parameters $\alpha$ and $\beta$.

| Dataset | $(\alpha, \beta) = (0.5, 0)$ | $(\alpha, \beta) = (0, 0.5)$ | $(\alpha, \beta) = (0.5, 0.5)$ | $(\alpha, \beta) = (1, 1)$ |
|---|---|---|---|---|
| News20 | $85.08_{\pm 0.12}$ | $84.95_{\pm 0.07}$ | $\underline{85.19}_{\pm 0.35}$ | $85.09_{\pm 0.20}$ |
| Review50 | $74.34_{\pm 0.10}$ | $74.76_{\pm 0.08}$ | $\underline{74.87}_{\pm 0.17}$ | $74.77_{\pm 0.13}$ |

**Different values for $\lambda_r$ in the reconstruction loss.** We conduct additional experiments with different values of $\lambda_r$ although we use a fixed value ($\lambda_r = 0.05$) for the reconstruction loss (Eq. 5) in all the experiments. As shown in Table 15, the reconstruction loss with moderate values of $\lambda_r$ clearly improves the performance, but too large value decreases the performance as it excessively updates the model for reconstruction loss. Hence, we recommend using the originally fixed value ($\lambda_r = 0.05$).

Table 15: Effect of different values for hyper-parameters $\lambda_r$.

| Dataset | $\lambda_r = 0$ | $\lambda_r = 0.05$ | $\lambda_r = 0.1$ | $\lambda_r = 0.5$ |
|---|---|---|---|---|
| News20 | $84.88_{\pm 0.19}$ | $\underline{85.19}_{\pm 0.35}$ | $85.10_{\pm 0.05}$ | $84.52_{\pm 0.23}$ |
| Review50 | $74.52_{\pm 0.03}$ | $\underline{74.87}_{\pm 0.17}$ | $74.89_{\pm 0.02}$ | $74.51_{\pm 0.17}$ |

**Update frequency $T_p$ and weight $\lambda_s$ for policy $\mathcal{T}_\phi$.** As described in Section 4.1, we mainly tune $T_p$ and $\lambda_s$ for each task. Table 16 and Table 17 show the results of different values of them. Here, we observe the consistent improvement of DND with varied $T_p$, which indicates an efficiency of DND. In the case of $\lambda_s$, the performance of DND still significantly outperforms the other baselines with all $\lambda_s$, but the gain is varied with the chosen $\lambda_s$, as this hyper-parameter directly affects the trade-off between 'informativeness' and 'risk' of the augmented samples.

Table 16: Effect of different $T_p$.

| Dataset | $T_p = 1$ | $T_p = 5$ | $T_p = 10$ |
|---|---|---|---|
| News20 | $84.99_{\pm 0.07}$ | $\underline{85.19}_{\pm 0.35}$ | $85.11_{\pm 0.15}$ |
| Review50 | $74.72_{\pm 0.22}$ | $\underline{74.87}_{\pm 0.17}$ | $74.89_{\pm 0.20}$ |

Table 17: Effect of different $\lambda_s$.

| Dataset | $\lambda_s = 0.1$ | $\lambda_s = 0.5$ | $\lambda_s = 1.0$ |
|---|---|---|---|
| News20 | $85.06_{\pm 0.27}$ | $\underline{85.19}_{\pm 0.35}$ | $84.97_{\pm 0.15}$ |
| Review50 | $74.17_{\pm 0.03}$ | $\underline{74.87}_{\pm 0.17}$ | $74.35_{\pm 0.04}$ |

**Effect of auxiliary operation.** As we have mentioned in Appendix A.3, we adopt the auxiliary operation for our implementation, which has been originally used by Hataya et al. (2020). Here, we provide the results without this auxiliary operation. As one can see in Table 18, this operation is not significant to the gain from DND.

Table 18: Effect of auxiliary operation (Hataya et al., 2020).

| Dataset | DND (no auxiliary) | DND |
|---|---|---|
| News20 | $85.14_{\pm 0.09}$ | $\underline{85.19}_{\pm 0.35}$ |
| Review50 | $74.79_{\pm 0.07}$ | $\underline{74.87}_{\pm 0.17}$ |

