# OpenReview forum: "What Makes Better Augmentation Strategies? Augment Difficult but Not too Different"
_ICLR.cc/2022/Conference — ICLR 2022 Poster_

### Official Review · Reviewer_PytR · 2021-11-02

**Correctness:** 4
**Technical Novelty And Significance:** 4
**Empirical Novelty And Significance:** 4
**Recommendation:** 8
**Confidence:** 4

**Main Review:**

##########################################################################
Strengths:
- The paper is very well written. The motivations and contributions of the paper are very clear and important. Data augmentation policy learning is a relatively under-explored area in the literature, and I think this paper makes a very useful contribution to exploring this research area. The proposed approach is simple enough to be replicated easily in follow-up research. The provided code in the supplementary material is well written and easy to understand too.
- The paper provides comprehensive experiments covering a diverse set of tasks. It includes results using 8 baseline/state-of-the-art models on 6 datasets with different text classification tasks, and on 8 tasks present in the GLUE benchmark. It provides the results on multiple class-imbalance/low-data setups. Further, it provides an ablation study to understand the contribution of different components of the proposed model, to provide evidence of the importance of each of those components. Finally, the paper also provides some results to show the transferability of the approach, and some analysis of learning dynamics. In most of these results, the proposed approach outperforms the baseline/state-of-the-art models.

##########################################################################
Weaknesses / Suggestions:
- Minor correction in page 4: "Rewarding not too different samples" section. It's mentioned to see "Figure 1(b) and 6(b)". But the figure 6 is not present in the main paper. Further, both "1(b)" and "6(b)" are linked to 1(b) figure only.
- `gW` from equation (3) is sent directly to the cross entropy loss function in equation (4). I think it would be more clear if a sigmoid function is added on `gW` either in equation (3) or (4). Further, in equation (3), did you consider simply calculating the dot product of `m1` and `m2`, instead of concatenating them and using a neural network?
- I think the equation (5) is a bit unclear. Could you clarify the following:
    - Is the idea here similar to minimizing the dot product between the output embedding and input embedding for a given word?
    - Can you add more description of what exactly `V` and `x` are? `x` is referred to as the input token, but its used as a label in the cross entropy loss, implying that it’s a binary value. How is the input token converted to a binary label?
    - Given that the output and input embeddings of a transformer are in different spaces, could you provide some justification of how calculating the dot product between those vectors would be appropriate?
- A minor comment, but the usage of policy/reward made me think that this paper uses some reinforcement learning approach. Upon reading the paper, the usage of those terms do seem appropriate. I am not suggesting that this needs a change, but I just wanted to point out that some readers might get confused a bit by this terminology.
- Regarding the GLUE benchmark results, it would be interesting to see how this model fares in the GLUE leaderboard. Could you add those results if available?
- In the final conclusion section, could you add some directions in which this work can be extended?

**Summary Of The Paper:**

The paper provides a simple yet effective approach towards data augmentation in NLP tasks. In particular, it proposes an augmentation strategy that encourages constructing augmented samples with low confidence (which makes them challenging, hence more informative) and high semantic similarity with the original samples (which ensures the semantics are not lost in augmentation). Through comprehensive experiments, the authors show that their approach outperforms recent state-of-the-art techniques, especially on low-data and class-imbalanced regimes.

**Summary Of The Review:**

Overall, I vote for accepting. I think the paper proposes a very interesting approach to improve data augmentation in NLP tasks by learning a policy that decides how to combine different augmentation techniques in a task dependent manner to generate samples that are informative while retaining the semantic information from the original sample. Since this research area is relatively new, I think this paper provides a really good baseline, and a framework for other papers to improve upon, as its easy to implement and allows adding more augmentation techniques in the policy pool, improving the loss/regularizer functions, etc. I have a few minor issues regarding the paper, and I described them in detail below. Hopefully the authors can address my concerns in the rebuttal period.

---

> ### Author Response · Authors · 2021-11-19
> **Response to Reviewer PytR (2/2)**
>
> **[C4] Can you add more description of what exactly $\mathbf{V}$ and $x$ are?**
>
> **[A4]** We present a more detailed description about $\mathbf{V}$ and $x$. Here, input token $x \in \\{ 0,1 \\} ^{V}$ is a $V$-dimensional one-hot vector with a vocabulary size $V$. $\mathbf{V} \in \mathbb{R}^{V \times d}$ is the word embeddings matrix of pre-trained transformer model that maps $x$ to $d$-dimensional input embedding $z \in \mathbb{R}^{d}$. This $z$ is feed-forwarded to the remaining parts of the transformer (e.g., self-attention blocks) and the corresponding output embedding is $o \in \mathbb{R}^{d}$. Hence, the resulting word reconstruction loss can be formulated as $\ell_{\tt xe}(p_{\mathbf{V}}(o), x)$ (equation (5)) where $p_{\mathbf{V}}(o) = \text{Softmax}(o\mathbf{V}^{\top})$, i.e., predicting a one-hot label (given by input token $x$) from the dot product of output embeddings $o$ and word embedding matrix $\mathbf{V}$. We have added these details in Appendix F of the revised manuscript.
>
> ---
>
> **[C5] Given that the output and input embeddings of a transformer are in different spaces, could you provide some justification of how calculating the dot product between those vectors would be appropriate?**
>
> **[A5]**  We would like to clarify that the output and the input embeddings are in the same space during the pre-training stage (e.g., masked token prediction in BERT or left-to-right token prediction in GPT). In such pre-training schemes, a dot product operation is used to reconstruct the original word tokens from the output embeddings. Using this dot product and the reconstruction loss allows to preserve the information within output embeddings learned through the pre-training phase; such a regularization has shown better knowledge transfer between the original and the target task [Gururangan et al., 2020].
>
> ---
>
> **[C6] Some readers might get confused a bit by this terminology (policy/reward)**
>
> **[A6]** Thank you for making constructive comments to improve the clarity of our manuscript. To prevent such confusion from the terminology, we have added the following comments on the footnote in Section 3.1 of the revised manuscript:
> - We use the terminologies of policy and reward following previous works [Li et al., 2020; Hataya et al., 2020], although our approach is not exactly a reinforcement learning algorithm.
>
> ---
>
> **[C7] Add some directions in which this work can be extended**
>
> **[A7]**  Following your suggestion, we have added possible directions in which our work can be extended in the revised manuscript as follow:
> - Furthermore, since the proposed concept of difficult but not too different augmentation is task- and domain-agnostic, we do believe that DND would benefit in other tasks (e.g., self-supervised learning) and domains (e.g., vision and graph) as well. On the other hand, incorporating the pre-trained models for augmentation learning in different domains (e.g., SimCLR [Chen et al., 2020]) would be an interesting future direction, as the fine-tuning of such pre-trained models have shown remarkable successes recently.
>
> ---
>
> **[C8] Results on GLUE leaderboard**
>
> **[A8]**  As you suggested, we believe the additional results on the GLUE leaderboard would greatly strengthen our paper. Due to limited time and resources, we could not update the results yet. Nevertheless, we will continue to work and deliver their results in the final revision (if time permits) or the final draft.
>
> ---
>
> **[C9] Minor correction in page 4**
>
> **[A9]**  Many thanks for the careful reading, especially for pointing out an error of link on page 4 to Figure 6(b) in Appendix B. In the revised manuscript, we have corrected this error.
>
> ---
> [Chen et al., 2020] A Simple Framework for Contrastive Learning of Visual Representations., ICML 2020
>
> [Gururangan et al., 2020] Don’t Stop Pretraining: Adapt Language Models to Domains and Tasks., ACL 2020
>
> [Li et al., 2020] DADA: Differentiable Automatic Data Augmentation., ECCV 2020
>
> [Hataya et al., 2020] Faster Autoaugment: Learning Augmentation Strategies using Backpropagation., ECCV 2020

---

> ### Author Response · Authors · 2021-11-20
> **Response to Reviewer PytR (1/2)**
>
> We sincerely appreciate your efforts and insightful comments to improve the manuscript. We respond to each of your comments one-by-one in what follows. In the revised manuscript, we have marked the revisions with “green.”  In tables, all the values and error bars are mean and standard deviation across 3 random seeds.
>
> ---
>
> **[C1] I think it would be more clear if a sigmoid function is added on $g_W$ either in equation (3) or (4)**
>
> **[A1]** Thank you for making constructive suggestions to improve the clarity of our manuscript. Following your comments, we explicitly describe the softmax function in equation (4) of the revised manuscript, which was inserted in the cross-entropy function in our previous manuscript.
>
> - Previous:  $\ell_{\tt xe} (g_W(m, \widetilde{m}), y_{\tt pos})$
>
> - Revised:  $\ell_{\tt xe}( p_W(m, \widetilde{m}), y_{\tt pos})$ where $p_{W}(m, \widetilde{m})= \text{Softmax} (g_{W}(m, \widetilde{m}))$
>
> ---
>
> **[C2] In equation (3), did you consider simply calculating the dot product of $m_1$ and $m_2$, instead of concatenating them and using a neural network?**
>
>  **[A2]** Following your suggestion (and reviewer A1ft), we conduct the additional experiments with DND using 1) cosine similarity and 2) ℓ2 distance between $m_1$ and $m_2$, instead of using MLP. As shown in the table below, the performance with the simpler choices decreases compared to the originally proposed design.
>
> | Dataset  |  Cosine similarity | ℓ2 distance | MLP |
> |:--------|:------------:|:----:|:------------:|
> | News20       | $84.42 \small{\pm 0.10}$ | $83.99 \small{\pm 0.29}$ | $\bf 85.19 \small{\pm 0.35}$ |
> | Review50    | $74.04 \small{\pm 0.07}$ | $73.81 \small{\pm 0.17}$ | $\bf 74.87 \small{\pm 0.17}$ |
>
>   Here, one can observe how using MLP is better than simpler choices (cosine-similarity or l2 distance). We believe that using the MLP allows a better tradeoff between the original task performance and learning the similarity between original and augmented sentences. Namely, the simpler choices (cosine-similarity or ℓ2 distance) could over-regularize the output embeddings and result in an underfitting for the target task. However, by leveraging MLP, more information for the target task can be formed and maintained in the output embeddings.
>
>   Finally, we remark that the effectiveness of MLP for measuring the similarity on the embeddings has been demonstrated in the other context as well, e.g., SimCLR [Chen et al., 2020]. We have added the respective discussions and results in Section 3.2 and Appendix F of the revised manuscript.
>
> ---
>
> **[C3] Is the idea here (Eq. 5) similar to minimizing the dot product between the output embedding and input embedding for a given word?"**
>
>  **[A3]** We clarify that the idea of equation (5) is not minimizing the dot product between the output embeddings and input embeddings, but predicting the original tokens from the output embeddings using a word embedding matrix, similar to pre-training loss used in the language model such as BERT. Hence, our reconstruction loss effectively preserves the information within output embeddings learned through the pre-training phase, and it could provide a better output embedding.
>
>   We empirically observed that the word reconstruction (i.e., prediction) from output embeddings becomes impossible as the model is fine-tuned in the target task (See added Figure 9 in Appendix F of the revised manuscript). We remark that this word reconstruction was almost perfectly possible before the fine-tuning, as the language model is pre-trained using a similar reconstruction loss (e.g., masked token prediction in BERT). Thus, this observation implies that this semantic information is lost. Our reconstruction loss effectively prevents such information loss and improves the quality of output embeddings.
>
>   To empirically demonstrate the effectiveness of this term, we further provide the experimental results of DND without reconstruction loss (i.e., $\lambda_r=0$). As shown in the below table, the reconstruction loss clearly improves the performance of DND by providing better output embeddings. We have added this discussion and results in Section 3.2. and Appendix F of the revised manuscript.
>
> | Dataset  | $\lambda_r=0$ | $\bf \lambda_r=0.05$ |
> |:--------|:------------:|:------------:|
> | News20       | $84.88 \small{\pm 0.19}$  | $\bf 85.19 \small{\pm 0.35}$ |
> | Review50    | $74.52 \small{\pm 0.03}$  | $\bf 74.87 \small{\pm 0.17}$ |
>
>   Lastly, we note that our initial manuscript could indicate some confusion regarding this, as the used $\ell_{\tt xe}$ is considered to include the softmax operation within itself. To clarify this, we revised the corresponding parts by specifying the softmax operation (equation (5) in Section 3.2).

---

### Official Review · Reviewer_A1ft · 2021-11-02

**Correctness:** 3
**Technical Novelty And Significance:** 2
**Empirical Novelty And Significance:** 3
**Recommendation:** 6
**Confidence:** 5

**Main Review:**

Overall, the paper is well-written and easy to understand. Experiments are conducted on different datasets and training schemes. The proposed method improves upon existing data augmentation techniques.

The proposed algorithm in this paper is a combination of known techniques, i.e., all the data augmentation techniques are well-established, and the relaxation method used to train the policy is also not new. Because the technical novelty of this paper is limited (which is acceptable), empirical evaluation needs to be substantially strengthened.

Some designs are not well-motivated.
1. In the similarity loss (Eq. 4), a two layer MLP is used to serve as a linear classifier. Could the authors explain why this is better than simpler choices such as cosine-similarity or $\ell_2$ distance?
2. Continue on Eq. 4, what is the accuracy of $g_W$ after training? It is possible that $g_W$ cannot get properly trained because of the limited number of fine-tuning epochs.
3. The reconstruction loss (Eq. 5) is confusing. Could the authors provide more intuition on this? Also, there should be experiments that demonstrate the necessity of this loss term.

The training looks ad-hoc. There are several designs that seem arbitrary. Some additional experiments are needed.
1. Several existing methods are selected as the data augmentation pool (Cutoff, Back-trans, BERT-aug, Adv, EDA and R3F). There should be experiments that show the importance of these policies, i.e., are they all necessary? In Figure 3a, the probability of selecting R3F, Adv, EDA and Cutoff is low (Cutoff nearly drops to 0) towards the end of training. Is it possible to only include Back-trans and BERT-aug?
2. The authors mention that they need to simultaneously train 4 augmentation policies. What will happen if we only train one (or more than 4) policy?
3. The operation count is set to $T=2$ in the experiments. Analysis are needed on other values of $T$.
4. The authors mention an auxiliary operation (Hataya et al., 2020) is added. An ablation study is needed regarding the significance of this operation.
5. In the sample weight $w$ (Eq. 1), there are two hyper-parameters $\alpha$ and $\beta$. Experiments are needed to see how these two hyper-parameters change model performance.
6. There should be experiments on the weight ($\lambda_r$) of the reconstruction loss (Algorithm 1).

Additional comments:
1. In Table 1, performance gain is not clear. Variance of the results is very large, and it is hard to tell the significance of the gain. The authors should conduct significance tests and report the p-values.
2. In the current version, the training loss function is only shown in the Algorithm box. Include it at the beginning of Section 3.2 will make the presentation clearer.

I will raise my score if the authors can conduct the ablation experiments and address my concerns.

**Summary Of The Paper:**

This paper proposes a data augmentation technique --- difficult but not too different. The authors claim that an augmented sample should be difficult (in terms of loss) and semantically similar to the original sample. The authors propose two rewards functions to reward difficult samples and similar samples, respectively. A relaxation technique is further used such that the augmentation policy can be trained. Experiments are conducted on the GLUE development set and several other text classification tasks. The proposed method improves upon existing ones.

**Summary Of The Review:**

The paper applies existing differentiable data augmentation methods (Hataya et al., 2020) to text classification. Overall, the technical novelty is low. There are several additional experiments (in particular ablation studies) needed to justify the design choices and the performance gain.

---

> ### Author Response · Authors · 2021-11-20
> **Response to Reviewer A1ft (3/3)**
>
> **[C6] Additional experiments about several designs (continued)**
>
> - *Number of operations $T$*: we compare the original choice ($T=2$) with less ($T=1$) and more ($T=3,4$) number of operations. Here, one can verify that a single operation is not enough to construct complexly augmented samples, hence the empirical gain is relatively small compared to the original choice. With more operations, almost saturated improvement is observed. Remarkably, similar tendencies are also observed in vision tasks [Hataya et al., 2020; Cubuk et al., 2020].
>
> | Dataset  |  $T=1$  | $T=2$ | $T=3$  | $T=4$ |
> |:--------|:------------:|:----:|:------------:|:----:|
> | News20       | $84.68 \small{\pm 0.41}$ | $\underline{85.19 \small{\pm 0.35}}$ | $85.19 \small{\pm 0.25}$ | $85.22 \small{\pm 0.10}$ |
> | Review50    | $74.24 \small{\pm 0.07}$ | $\underline{74.87 \small{\pm 0.17}}$ | $74.93 \small{\pm 0.07}$ | $75.01 \small{\pm 0.02}$ |
>
> - *Significance of auxiliary operation*: here, we provide the results without an auxiliary operation [Hataya et al., 2020]. As one can see, this operation is not significant to the gain from DND.
>
> | Dataset  |   DND (no auxiliary operation)   | DND |
> |:--------|:------------:|:----:|
> | News20       | $85.14 \small{\pm 0.09}$ | $\underline{85.19 \small{\pm 0.35}}$ |
> | Review50    | $74.79 \small{\pm 0.07}$ | $\underline{74.87 \small{\pm 0.17}}$ |
>
> - *Ablation for hyper-parameters $\alpha$ and $\beta$*: we additionally provide the results with different values for $\alpha$ and $\beta$ in the below tables. Here, it is observable that each component of the re-weighting scheme is solely effective and the gain from re-weighting is not much sensitive to the choice of $\alpha$ and $\beta$. However, we remark the original choice $(\alpha, \beta)=(0.5,0.5)$ shows the best empirical gains.
>
> | Dataset  |  $(\alpha, \beta)=(0.5,0)$  | $(\alpha, \beta)=(0,0.5)$ | $(\alpha, \beta)=(1,1)$  | $(\alpha, \beta)=(0.5,0.5)$ |
> |:--------|:------------:|:----:|:------------:|:----:|
> | News20       | $85.08 \small{\pm 0.12}$ | $84.95 \small{\pm 0.07}$ | $85.09 \small{\pm 0.20}$ | $\underline{85.19 \small{\pm 0.35}}$ |
> | Review50    | $74.34 \small{\pm 0.10}$ | $74.76 \small{\pm 0.08}$ | $74.77 \small{\pm 0.13}$ | $\underline{74.87 \small{\pm 0.17}}$ |
>
> - *Weight ($\lambda_r$) for the reconstruction loss*: following your suggestion, we conduct additional experiments with different values of $\lambda_r$ although we use a fixed value ($\lambda_r=0.05$) in all the experiments. As shown in the below table, the reconstruction loss with moderate values of $\lambda_r$ clearly improves the performance, but too large value decreases the performance as it excessively updates the model for reconstruction loss. Hence, we recommend using the originally fixed value ($\lambda_r=0.05$).
>
> | Dataset  | $\lambda_r=0$ | $\lambda_r=0.05$ | $ \lambda_r=0.1$ | $\lambda_r=0.5$  |
> |:--------|:------------:|:----:|:----:|:------------:|
> | News20       | $84.88 \small{\pm 0.19}$  | $\underline{85.19 \small{\pm 0.35}}$ | $ 85.10 \small{\pm 0.05}$ | $84.52 \small{\pm 0.23}$ |
> | Review50    | $74.52 \small{\pm 0.08}$  | $\underline{74.87 \small{\pm 0.17}}$ | $ 74.89 \small{\pm 0.02}$ | $74.51 \small{\pm 0.17}$ |
>
> ---
>
> **[C7] The authors should conduct significance tests and report the p-values**
>
> **[A7]** Following your suggestion, we perform t-test on the results in Tables 1 and 2 (which show relatively large variance) to validate the statistical significance of DND compared to other baselines. Here, we observe that the average p-value is 0.050, which is generally known to indicate the statistical significance of the results. To clearly show the performance gain from DND, we have added this result in Section 4.2 of the revised manuscript.
>
> ---
>
> **[C8] In the current version, the training loss function is only shown in the Algorithm box**
>
> **[A8]** Thank you for making constructive suggestions to improve the clarity of our manuscript. We have included the overall training loss in Eq. 6 at the end of Section 3.2 for a clearer presentation in the revised manuscript.
>
> ---
>
> [Zhang et al., 2020] Adversarial Autoaugment., ICLR 2020
>
> [Hataya et al., 2020] Faster Autoaugment: Learning Augmentation Strategies using Backpropagation., ECCV 2020
>
> [Chen et al., 2020] A Simple Framework for Contrastive Learning of Visual Representations., ICML 2020

---

> ### Author Response · Authors · 2021-11-20
> **Response to Reviewer A1ft (2/3)**
>
> **[C4] The reconstruction loss (Eq. 5) is confusing. Could the authors provide more intuition on this? Also, there should be experiments that demonstrate the necessity of this loss term.**
>
> **[A4]** Our reconstruction loss (Eq. 5) allows the output embedding to preserve its knowledge learned from the pre-training phase, i.e., its ability to reconstruct words from sentences. Without such a regularization, the word reconstruction accuracy degrades from almost perfect to nearly impossible during the fine-tuning stage (See added Figure 9 in Appendix F of the revised manuscript). Remarkably, this regularization has already shown good performance for transferring knowledge between the pre-training and the target task [Gururangan et al., 2020].
>
>   To empirically demonstrate the effectiveness of this term, we further provide the experimental results of DND without reconstruction loss (i.e., $\lambda_r=0$). As shown in the below table, the reconstruction loss clearly improves the performance of DND by providing better output embeddings. We have added this discussion and results in Section 3.2. and Appendix F of the revised manuscript.
>
> | Dataset  | $\lambda_r=0$ | $\bf \lambda_r=0.05$|
> |:--------|:------------:|:------------:|
> | News20       | $84.88 \small{\pm 0.19}$  | $\bf 85.19 \small{\pm 0.35}$ |
> | Review50    | $74.62 \small{\pm 0.08}$  | $\bf 74.87 \small{\pm 0.17}$ |
>
>   Lastly, we note that our initial manuscript could indicate some confusion regarding this, as the used $\ell_{\tt xe}$ is considered to include the softmax operation within itself. To clarify this, we revised the corresponding parts by specifying the softmax operation (Eq. 5 in Section 3.2).
>
> ---
>
> **[C5] Several existing methods are selected as the data augmentation pool ... There should be experiments that show the importance of these policies, i.e., are they all necessary?**
>
> **[A5]** To alleviate your concern, we conducted additional experiments with a reduced augmentation pool including only the word-level augmentations (Back-trans, BERT-aug, and EDA) for the first operation of DND.
>
> | Dataset  | Baseline | DND (reduced) | DND |
> |:--------|:------------:|:----:|:----:|
> | News20       | $83.37 \small{\pm 0.23}$ | $84.99 \small{\pm 0.29}$ | $\bf 85.19 \small{\pm 0.35}$ |
> | Review50    | $73.34 \small{\pm 0.11}$ | $74.81 \small{\pm 0.01}$ | $\bf 74.87 \small{\pm 0.17}$ |
>
>   As shown in the above table, the performance of DND is slightly decreased with the reduced augmentation pool, but it still outperforms other baseline augmentations (in the second column, we report the best performance among baselines for your convenience). This result demonstrates the advantage of augmentation learning frameworks again. Also, it might indicate the exploration among the large augmentation pool at the early stage improves the generalization of the model, although some augmentations have low probabilities at the end (as shown in Figure 3 (a)).
>
>   Finally, we remark that our main contribution lies in the algorithm to learn the augmentation policy, rather than proposing a new data augmentation pool. Even when our pool includes augmentation methods useless for the target task, our DND will be able to filter out those augmentations; this is the main advantage of using learning-based over non-learning-based data augmentation algorithms. We believe our algorithm can be easily improved by considering other choices of data augmentation pool, and we leave this for future work.
>
> ---
>
> **[C6] Additional experiments about several designs**
>
> **[A6]** Following your suggestion, we conduct additional experiments about several designs of DND as follows. Here, the original design choices of DND are underlined. In summary, these results indicate that DND is not sensitive to the choice of hyper-parameters. All the respective discussions and results are added in Appendix G of the revised draft.
> - *Number of augmentation policies ($n_{\mathcal{T}}$)*: we compare the original choice ($n_{\mathcal{T}}=4$) with less ($n_{\mathcal{T}}=1$) and more ($n_{\mathcal{T}}=10$) number of policies. Overall, the improvement from DND is not sensitive to the number of augmentation policies, but using multiple policies shows slightly better results as it would construct more diverse augmentations.
>
> | Dataset  |  $n_{\tt policy}=1$  | $n_{\tt policy}=4$  | $n_{\tt policy}=10$  |
> |:--------|:------------:|:----:|:------------:|
> | News20       | $84.93 \small{\pm 0.05}$ | $\underline{85.19 \small{\pm 0.35}}$ | $85.16 \small{\pm 0.08}$ |
> | Review50    | $74.84 \small{\pm 0.18}$ | $\underline{74.87 \small{\pm 0.17}}$ | $74.90 \small{\pm 0.17}$ |

---

> ### Author Response · Authors · 2021-11-20
> **Response to Reviewer A1ft (1/3)**
>
> We sincerely appreciate your efforts and insightful comments to improve the manuscript. We respond to each of your comments one-by-one in what follows. In the revised manuscript, we have marked the revisions with “green.” In tables, all the values and error bars are mean and standard deviation across 3 random seeds.
>
> ---
>
> **[C1] The paper applies existing differentiable data augmentation methods [Hataya et al., 2020] to text classification. Overall, the technical novelty is low.**
>
> **[A1]** We first clarify that our core contribution is proposing a new reward function by exploring a novel combination of difficulty and similarity for effective augmentation policy learning, rather than applying differentiable augmentation [Hataya et al., 2020] to NLP tasks. We remark that such a combination of two aspects has not been explored before, although each of difficulty [Zhang et al., 2020] and similarity [Hataya et al., 2020] has been independently investigated. Especially, as the problem is relatively under-explored in NLP tasks, designing practical objectives for achieving these rewards is highly non-trivial.
>
>   Also, we propose new ideas of sample-wise adaptation and leverage of pre-training model for designing novel reward functions, different to the previous works using sample-agnostic reward with additional external models.
> - Sample-wise adaptation for the augmentation learning is a promising yet under-explored way, and our proposed sample-wise re-weighting scheme would be a valuable contribution in this direction.
> - For the "Similarity" side, on the other hand, we suggest a new idea of leveraging the information from the language model itself; hence it can reduce the cost from designing external modeling for this, such as GAN [Hataya et al., 2020]. Also, this idea would reveal a new interesting direction of incorporating the pre-trained models for augmentation learning in other domains (e.g., SimCLR) as well.
>
>
>   Overall, data augmentation policy learning is a relatively under-explored area in the literature and we do believe that our work makes a “very useful contribution” (as highlighted by Reviewer PytR) to exploring this research area.
>
> ---
>
> **[C2] Could the authors explain why this is better than simpler choices such as cosine-similarity or ℓ2 distance?**
>
> **[A2]** We believe that using the MLP allows a better tradeoff between the original task performance and learning the similarity between original and augmented sentences. Namely, the simpler choices (cosine-similarity or ℓ2 distance) could over-regularize the output embedding and result in an underfitting for the target task. However, by leveraging the expressive power of MLP, more information for the target task can be formed and maintained in the output embeddings.
>
>   To empirically validate this intuition, we conduct the additional experiments with DND using 1) cosine similarity and 2) ℓ2 distance between $m_1$ and $m_2$, instead of using MLP. As shown in the table below, the performance with the simpler choices actually decreases compared to using MLP.
>
> | Dataset  |  Cosine similarity | ℓ2 distance | MLP |
> |:--------|:------------:|:----:|:------------:|
> | News20       | $84.42 \small{\pm 0.10}$ | $83.99 \small{\pm 0.29}$ | $\bf 85.19 \small{\pm 0.35}$ |
> | Review50    | $74.04 \small{\pm 0.07}$ | $73.81 \small{\pm 0.17}$ | $\bf 74.87 \small{\pm 0.17}$ |
>
>   Finally, we remark that the effectiveness of MLP for measuring the similarity on the embeddings has been demonstrated in the other context as well, e.g., SimCLR [Chen et al., 2020]. We have added the respective discussions and results in Section 3.2 and Appendix F of the revised manuscript.
>
> ---
>
> **[C3] Continue on Eq. 4, what is the accuracy of gW after training? It is possible that gW cannot get properly trained because of the limited number of fine-tuning epochs.**
>
> **[A3]** To alleviate your comments, we have plotted the accuracy of $g_{W}$ in Figure 8 of Appendix F of the revised manuscript. In the experiments, we find that $g_{W}$ does get properly trained and easily achieves high accuracy, e.g., $g_{W}$ achieves 98.1% accuracy only after one epoch of training on the Review50 dataset. As mentioned in our previous response (**A2**), this is thanks to parameterizing $g_{W}$ that allows a better tradeoff between the original task and accuracy of $g_{W}$. We have incorporated a more comprehensive discussion and result in Appendix F of the revised manuscript.

---

> ### Comment · Reviewer_A1ft · 2021-11-21
> **Response to author rebuttal**
>
> I stand on my point that technical novelty is limited as the paper combines well-known methods. However, I also believe that technical novelty alone does not define quality of a paper. The initial version left out critical ablation experiments as the proposed algorithm introduces several hyper-parameters. The added experiments and modification to the paper has addressed my concerns.

---

> > ### Author Response · Authors · 2021-11-22
> > **Thank you for the response**
> >
> > Thank you for the positive response before the discussion phase ends, and we are happy to hear that our response could help to address your concerns !
> >
> > We also agree that we used well-known techniques optimizing the proposed reward function, but we still think that designing such a good reward function itself, in particular working well for NLP tasks, is challenging and novel.
> >
> > If you have any further questions or concerns, please do not hesitate to let us know.
> >
> > Thanks,
> >
> > Authors

---

### Official Review · Reviewer_SErr · 2021-11-07

**Correctness:** 3
**Technical Novelty And Significance:** 2
**Empirical Novelty And Significance:** 3
**Recommendation:** 6
**Confidence:** 3

**Main Review:**

Strengths:
The proposed method is intuitive, simple to apply and effective over many baseline augmentation methods on various datasets, especially on low-resource, class-imbalanced regimes. Besides the good performance on benchmarks, the authors also conducted good ablation study and analysis on learned policies, showing that the selected policy coincides with previous work (via exhaustive grid search).

Weakness and Concerns:
1. The main issue lies in the choice of baseline. As a learning-based augmentation method, it will be more convincing to compare with other learning-based method (using same pool), e.g. auto-augment, adversarial auto-augment etc. If the method can still outperform them, it can better demonstrate the effectiveness of their reward design.
2. Limited technical novelty: The whole idea is very straight-forward and most of the techniques are borrowed from previous works.

**Summary Of The Paper:**

The authors proposed a new reward function to learn the distribution of augmentation policies for NLP tasks, which can generate "difficult" and semantic similar samples to facilitate training. They further introduced a sample-wise re-weighting scheme to leverage the learning status of original sample. A continuous relaxation is applied to optimize the trainable parameters in policy. The proposed method outperformed SOTA on text classification and entailment tasks. They also conducted extensive studies and analysis, demonstrating the effectiveness of their method on low-resource and class-imbalanced regimes, as well as its transferability.

**Summary Of The Review:**

This paper proposed a simple and effective reward function for learning-based augmentation selection in NLP. Extensive experiments and analysis demonstrated its effectiveness on text classification and entailment, especially on low-resource and class-imbalanced regimes.

---

> ### Author Response · Authors · 2021-11-20
> **Response to Reviewer SErr (1/1)**
>
> We sincerely appreciate your efforts and insightful comments to improve the manuscript. We respond to each of your comments one-by-one in what follows. In the revised manuscript, we have marked the revisions with “green.” In tables, all the values and error bars are mean and standard deviation across 3 random seeds.
>
> ---
>
> **[C1] Compare with other learning-based method ... demonstrate the effectiveness of their reward design**
>
> **[A1]** We first remark that we have already compared DND to other learning-based methods with different reward designs in Table 4; for example, "Difficult" in Table 4 updates the augmentation policy to maximize the training loss, as like adversarial auto-augment. Hence, the results in Table 4 indicate the effectiveness of our reward design compared to that of adversarial auto-augment.
>
>   To further clarify your concern (and also requested by Reviewer XAnt), we additionally compare DND with another learning-based method with a different reward design, which directly maximizes the validation accuracy to update the augmentation policy as like auto-augment. Since auto-augment requires massive computations than the continuous relaxation method [Hataya et al., 2020], which is adopted to our method (e.g., 5000 vs. 0.23 GPU hours in CIFAR10 by [Hataya et al., 2020]), we instead consider LDM [Hu et al., 2019] that also maximizes the validation accuracy to learn the augmentation in a more efficient way, by adapting an off-the-shelf reward learning algorithm from RL. Since LDM originally uses a different augmentation pool, we modified LDM to use the same augmentation pool with DND. Also, we apply extra losses (Eq. 4 and Eq.5) with LDM as these losses slightly improve the overall performance, as shown in Appendix F of the revised draft.
>
>   As summarized below, we indeed observe that DND outperforms LDM. These results demonstrate that our reward design is more effective than maximizing the validation accuracy for learning the augmentations. Furthermore, we emphasize that maximizing validation accuracy usually requires more computations than training loss-based rewards due to an inherent bi-level optimization, as shown in [Zhang et al. 2020; Hataya et al., 2020]. For example, LDM is 3x slower than DND under the same update frequency in our experiments. We have added the discussion and the results of LDM in Section 4.3 and Table 4, respectively.
>
> | Dataset  |  LDM [Hu et al., 2019]  | DND |
> |:--------|:------------:|:----:|
> | News20       | $85.08 \small{\pm 0.07}$ | $\bf 85.19 \small{\pm 0.35}$ |
> | Review50    | $73.72 \small{\pm 0.22}$ | $\bf 74.87 \small{\pm 0.17}$ |
>
> ---
>
> **[C2] Limited technical novelty**
>
> **[A2]** We first emphasize that our core contribution is proposing a new reward function by exploring a novel combination of difficulty and similarity for effective augmentation policy learning. We remark that such a combination of two aspects has not been explored before, although each of difficulty [Zhang et al., 2020] and similarity [Hataya et al., 2020] has been independently investigated. Also, as this problem is relatively under-explored in NLP tasks, designing practical objectives for achieving these rewards is highly non-trivial.
>
>   Also, we propose new ideas of sample-wise adaptation and leverage of pre-trained models for designing novel reward functions, different to the previous works using sample-agnostic reward with additional external models.
>
> - Sample-wise adaptation for the augmentation learning is a promising yet under-explored way, and our proposed sample-wise re-weighting scheme would be a valuable contribution in this direction.
> - For the "Similarity" side, on the other hand, we suggest a new idea of leveraging the information from the language model itself; hence it can reduce the cost from designing external modeling for this, such as GAN [Hataya et al., 2020]. Also, this idea would reveal a new interesting direction of incorporating the pre-trained models for augmentation learning in other domains (e.g., SimCLR) as well.
>
>
>   Overall, data augmentation policy learning is a relatively under-explored area in the literature and we do believe that our work makes a “very useful contribution” (as highlighted by Reviewer PytR) to exploring this research area.
>
> ---
>
> [Hu et al. 2019] Learning Data Manipulation for Augmentation and Weighting., NeurIPS 2019
>
> [Zhang et al., 2020] Adversarial Autoaugment., ICLR 2020
>
> [Hataya et al., 2020] Faster Autoaugment: Learning Augmentation Strategies using Backpropagation., ECCV 2020

---

### Official Review · Reviewer_XAnt · 2021-11-07

**Correctness:** 3
**Technical Novelty And Significance:** 2
**Empirical Novelty And Significance:** 3
**Recommendation:** 6
**Confidence:** 4

**Main Review:**

Strengths:
-	__Clarity__. The writing is clear and easy to follow.
-	__Well-motivated approach with extensive experiments.__ Although the approaches to learn an augmentation policy using an adversarial objective, enforce the augmented samples to be similar to the original samples and use continuous relaxation to train the augmentation policy are studied separately in image domain, the proposed method is a viable way to apply the ideas to text data and shows good experimental results on various NLP tasks.

Weaknesses/comments:
-	__Comparison with other learning objectives.__ The ultimate goal of data augmentation is to improve the generalization power of a model. How does the proposed Difficult and Not Different objective compare with the objective that improves the validation performance directly like AutoAugment or TAA [1].
-	__Sensitivity of DND hyperparameters.__ Despite being an Automated Data Augmentation method, there are several important hyperparameters, e.g. $\lambda_r, \lambda_{sim}, \lambda_s, T_p$, to be tuned. How sensitive is the proposed method to these hyperparameters? Whether these hyperparameters needed to be tuned carefully?
-	__Ablation study for extra loss terms.__ In the ablation study section, are the ‘Vanilla’, ‘Random’ and ‘Fixed’ baselines also being trained with the extra loss terms $L_{sim}$ and $L_{recon}$? While it is a motivated decision to introduce these terms during the training of $f_\theta$, can these losses also contribute to the learning of better representations and lead to the improvement? It would be useful if the effects of these extra losses are discussed in the ablation study.
-	__Exclusion of MixUp from the augmentation pool.__ Authors mention that MixUp is not included in the augmentation pool as it alters the semantics of the original sentences. However, MixUp can create difficult examples by sample and label mixing. MixUp also shows good results for some datasets under Table 1. As a learning-based data augmentation method, is it the responsibility for the search algorithm to learn the use of MixUp in a data-driven way? Can the inclusion of MixUp in the augmentation pool improves the performance? If the validity of a candidate augmentation method has to be evaluated before adding to the augmentation pool, does it contradict with the goal of fully learnable data augmentation?
-	__Hyperparameter tuning for other baselines.__ Some of the baseline methods, like EDA, BERT-Aug, MixUp, Back&Adv also involve hyperparameters. According to Appendix A2, it seems that these hyperparameters are not tuned. As the hyperparameters of DND (e.g. $\lambda_s$ and $T_p$) are tuned for each dataset, it is fairer to tune the hyperparameters for the other baselines.
-	__Formulation of the probability and magnitude for EDA.__  Is there a specific reason that EDA is assigned with a single probability and magnitude parameter? Can the use of different $p$ and $\mu$ values for synonym replacement, random insertion, random swap, and random deletion further improves the augmentation policy?
-	__Definition of the magnitude parameters for the augmentation candidates.__ It is unclear how the magnitude parameters are defined for the augmentation candidates. For example, is the mask probability of BERT-Aug taken as the magnitude? What are the ranges of the magnitudes? It would be useful to include these in the appendix.

Reference

[1] Shuhuai Ren, Jinchao Zhang, Lei Li, Xu Sun, Jie Zhou. Text AutoAugment: Learning Compositional Augmentation Policy for Text Classification. _arXiv preprint arXiv:2109.00523,_ 2021

**Summary Of The Paper:**

The proposed method DND learns an augmentation policy to augment a target text dataset. The augmentation policy is optimized to create difficult yet not too different examples from the original data. To further improve the performance, a sample re-weighting scheme is introduced to focus on harder examples. Experiments show that learned augmentation policy can achieve good results on various NLP tasks and transfer well across datasets and architectures.

**Summary Of The Review:**

Although the approach to learn an augmentation policy using the adversarial and similarity objectives is studied in the image domain, the proposed method contributes to the adaptation of learnable augmentation in the NLP domain and shows good experimental results. Overall, I tend to recommend for acceptance for the initial rating.

---

> ### Author Response · Authors · 2021-11-20
> **Response to Reviewer XAnt (4/4)**
>
> **[C7] Definition of the magnitude parameters for the augmentation candidates**
>
> **[A7]** The definition of the magnitude parameters for each augmentation candidate is as follow: 1) Adversarial: a magnitude for the norm of adversarial noise, 2) Cutoff: a length of the removed segment from the sentence, 3) R3F: a magnitude of the norm of random noise, 4) Back-trans: not exist, 5) BERT-aug: a masking probability from BERT model, and 6) EDA: a probability of applying one of EDA operations.
>
>   Here, although both BERT-aug and EDA have their own magnitude, we do not update these magnitudes, as it incurs the computational burden from the additional inference of external model (e.g., BERT) for each training iteration. Instead, we adopt the common practice in text augmentations [Wei and Zou, 2019; Yi et al., 2021; Qu et al., 2021], which generate the augmented sentences using the pre-determined magnitude before training the model, similar to the case of Back-trans. Hence, we only optimize the probability parameters for those word-level augmentations (BERT-aug, Back-trans, and EDA), as mentioned on page 6. However, as it expands the search space of DND at the expense of more computation, we agree that updating those magnitudes could give an additional improvement and leave this for future work. We have clarified these details and added the respective discussions in Section 4.1 and Appendix A.3 of the revised draft.
>
> ---
>
> [Hu et al. 2019] Learning Data Manipulation for Augmentation and Weighting., NeurIPS 2019
>
> [Wei and Zou, 2019] EDA: Easy Data Augmentation Techniques for Boosting Performance on Text Classification Tasks., EMNLP-IJCNLP 19
>
> [Yi et al., 2021] Reweighting Augmented Samples by Minimizing the Maximal Expected Loss, ICLR 21
>
> [Qu et al., 2021] Contrast-enhanced and Diversity-promoting Data Augmentation for Natural Language Understanding, ICLR 21

---

> ### Author Response · Authors · 2021-11-20
> **Response to Reviewer XAnt (3/4)**
>
> **[C3] Ablation study for extra loss terms (continued)**
>
> Also, to verify the effects of the extra losses for learning representation, we conduct the experiments using baseline augmentations (Adversarial, Cutoff, Back-trans) by applying these losses on News20 dataset. Here, when each loss is applied, the fixed coefficient used in DND is applied. Then, it can be observed that each of the proposed losses could improve the performance by contributing to the learning of better representations, and two extra losses are complementary to each other. We have added these results in Appendix F of the revised manuscript.
>
> | Dataset  |  $\mathcal{L}_{\tt sim}$  | $\mathcal{L}_{\tt recon}$  | Cutoff | Back-trans | Adversarial |
> |:--------|:------------:|:----:|:------------:|:------------:|:------------:|
> | News20       | - | - | $83.36 \small{\pm 0.41}$ |  $82.57 \small{\pm 0.60}$ |  $83.37 \small{\pm 0.23}$ |
> |    | $\checkmark$ | - | $83.74 \small{\pm 0.42}$ |  $82.77 \small{\pm 0.49}$ |  $83.63 \small{\pm 0.15}$ |
> |    | - | $\checkmark$ | $83.52 \small{\pm 0.13}$ |  $82.86 \small{\pm 0.47}$ |  $83.44 \small{\pm 0.08}$ |
> |    | $\checkmark$ | $\checkmark$ | $\bf 84.06 \small{\pm 0.19}$ |  $\bf 83.01 \small{\pm 0.23}$ |  $\bf 83.79 \small{\pm 0.56}$ |
>
> ---
>
> **[C4] Exclusion of MixUp from the augmentation pool**
>
> **[A4]** MixUp is the only exceptional case that we excluded from the augmentation pool. Typically, most augmentations assume that original and augmented samples are semantically similar, which is not the case for MixUp (it synthesizes a completely new sample by mixing two original samples). Hence, it is not clear whether it is meaningful to keep the similarity between original and augmented samples, which is the key component of DND. Except for MixUp, we think any other augmentations can be in our augmentation pool.
>
>  Nevertheless, following the suggestion made by you, we include MixUp in our augmentation pool by modifying the equation (2) and (4) similar to cross-entropy for MixUp as follow:
> - $\mathcal{R}{\tt sim} := -( \lambda * \ell{\tt xe}(g_{W}(m_1, \tilde{m}), y_{\tt pos}) + (1 - \lambda) * \ell_{\tt xe}(g_{W}(m_2, \tilde{m}), y_{\tt pos}))$
> - $\mathcal{L}{\tt sim} := \lambda * \ell{\tt xe}(g_{W}(m_1,\tilde{m}), y_{\tt pos}) + (1 - \lambda) * \ell_{\tt xe}(g_{W}(m_2, \tilde{m}), y_{\tt pos}) + \ell_{\tt xe}(g_{W}(m,n), y_{\tt neg}))$ where $\tilde{m}$ is the mean-pooled embeddings of augmented (mixed) sample $\tilde{x}= \lambda x_1 + (1-\lambda) x_2$.
>
> | Dataset  |  DND with MixUp  | DND |
> |:--------|:------------:|:----:|
> | News20       | $85.07 \small{\pm 0.10}$ | $\bf 85.19 \small{\pm 0.35}$ |
> | Review50    | $74.68 \small{\pm 0.07}$ | $\bf 74.87 \small{\pm 0.17}$ |
>
>   Here, we observe that DND with MixUp still outperforms other baselines, but the performance is slightly lower than the original DND. Such degradation might be because the considered modification is insufficient for MixUp, or it negatively affects measuring the similarity of other augmentation as we have been concerned initially. We have clarified it in Section 4.1 and added more discussions in Appendix A.3 of the revised manuscript.
>
> ---
>
> **[C5] Hyperparameter tuning for other baselines**
>
> **[A5]** Thank you for pointing out this detail for the fairer comparison. Following your suggestion, we additionally tuned the hyperparameters of EDA, BERT-Aug, MixUp, and Back&Adv as follow:
> - EDA; $p_{\tt eda} \in \\{0.15, 0.3, 0.45 \\}$
> - BERT-aug; $p_{\tt mask} \in \\{0.15, 0.3, 0.45 \\}$
> - MixUp; $\alpha_{\tt mix} \in \\{0.5, 1.0, 2.0 \\}$
> - Back&Adv; $\delta \in \\{0.1, 0.2, 0.3 \\}$
>
> Under these hyperparameters, we revised some experimental results of these baselines in Tables 1, 2, 3, 5, and 6 (the changed values are colored in "green") of the revised manuscript. We remark that DND still consistently outperforms all the baselines (the overall trend on the superiority of DND over baselines is the same as before). For example, with the revised results with full datasets in Table 1, the average relative test error reduction compared to previous best augmentation becomes 8.59% from 9.08%.
>
> ---
>
> **[C6] Formulation of the probability and magnitude for EDA**
>
> **[A6]** We assign a single parameter for EDA following the practice of the original paper [Wei and Zou, 2019]. Here, one common parameter has been shared across to augment the sentences, although there are four operations in EDA. Since we have focused on incorporating well-established EDA augmentation itself (i.e., randomized operation) into our augmentation pool rather than considering each operation, we also use the one common parameter for EDA. But, as the reviewer has mentioned, we agree that introducing separated parameters for each operation could improve the augmentation policy, and it would be an interesting future direction.

---

> ### Author Response · Authors · 2021-11-20
> **Response to Reviewer XAnt (2/4)**
>
> **[C2] Sensitivity of DND hyperparameters (continued)**
>
> - *Ablation for hyper-parameters $\alpha$ and $\beta$*: we additionally provide the results with different values for $\alpha$ and $\beta$ in the below tables. Here, it is observable that each component of the re-weighting scheme is solely effective and the gain from re-weighting is not much sensitive to the choice of $\alpha$ and $\beta$. However, we remark the original choice $(\alpha, \beta)=(0.5,0.5)$ shows the best empirical gains.
>
> | Dataset  |  $(\alpha, \beta)=(0.5,0)$  | $(\alpha, \beta)=(0,0.5)$ | $(\alpha, \beta)=(1,1)$  | $(\alpha, \beta)=(0.5,0.5)$ |
> |:--------|:------------:|:----:|:------------:|:----:|
> | News20       | $85.08 \small{\pm 0.12}$ | $84.95 \small{\pm 0.07}$ | $85.09 \small{\pm 0.20}$ | $\underline{85.19 \small{\pm 0.35}}$ |
> | Review50    | $74.34 \small{\pm 0.10}$ | $74.76 \small{\pm 0.08}$ | $74.77 \small{\pm 0.13}$ | $\underline{74.87 \small{\pm 0.17}}$ |
>
> - *Weight ($\lambda_r$) for the reconstruction loss*: following your suggestion, we conduct additional experiments with different values of $\lambda_r$ although we use a fixed value ($\lambda_r=0.05$) in all the experiments. As shown in the below table, the reconstruction loss with moderate values of $\lambda_r$ clearly improves the performance, but too large value decreases the performance as it excessively updates the model for reconstruction loss. Hence, we recommend using the originally fixed value ($\lambda_r=0.05$).
>
> | Dataset  | $\lambda_r=0$ | $ \lambda_r=0.05$ | $ \lambda_r=0.1$ | $\lambda_r=0.5$  |
> |:--------|:------------:|:----:|:----:|:------------:|
> | News20       | $84.88 \small{\pm 0.19}$  | $\underline{85.19 \small{\pm 0.35}}$ | $ 85.10 \small{\pm 0.05}$ | $84.52 \small{\pm 0.23}$ |
> | Review50    | $74.52 \small{\pm 0.08}$  | $\underline{74.87 \small{\pm 0.17}}$ | $ 74.89 \small{\pm 0.02}$ | $74.51 \small{\pm 0.17}$ |
>
> Furthermore, we emphasize that many hyper-parameters are actually used with a fixed value; for example, we use $\lambda_{r}=0.05$ for all experiments. Regarding the mainly tuned hyper-parameters $\lambda_s$ and $T_{p}$, we provide the additional experimental results by varying their values.
> - *Update frequency ($T_{p}$) and Weight ($\lambda_s$) for the policy*: here, we observe the consistent improvement of DND with varied $T_{p}$, which indicates an efficiency of DND. In the case of $\lambda_s$, the performance of DND still significantly outperforms the baselines with all $\lambda_s$, but the gain is varied with the chosen $\lambda_s$, as this hyper-parameter directly affects the trade-off between 'informativeness' and 'risk' of the augmented samples.
>
> | Dataset  |  $T_{p}=1$  | $T_{p}=5$ | $T_{p}=10$ |
> |:--------|:------------:|:----:|:------------:|
> | News20       | $84.99 \small{\pm 0.07}$ | $\underline{85.19 \small{\pm 0.35}}$ | $85.11 \small{\pm 0.15}$ |
> | Review50    | $74.72 \small{\pm 0.22}$ | $\underline{74.87 \small{\pm 0.17}}$ | $74.89 \small{\pm 0.20}$ |
>
> | Dataset  |  $\lambda_{s}=0.1$  | $\lambda_{s}=0.5$ | $\lambda_{s}=1.0$ |
> |:--------|:------------:|:----:|:------------:|
> | News20       | $85.06 \small{\pm 0.27}$ | $\underline{85.19 \small{\pm 0.35}}$ | $84.97 \small{\pm 0.15}$ |
> | Review50    | $74.17 \small{\pm 0.03}$ | $\underline{74.87 \small{\pm 0.17}}$ | $74.35 \small{\pm 0.04}$ |
>
> ---
>
> **[C3] Ablation study for extra loss terms**
>
> **[A3]** While 'Difficult' and 'Not Different' are trained with the extra loss terms $\mathcal{L}_{\tt sim}$ and $\mathcal{L}_{\tt recon}$, other methods are not being trained with them now. To clarify the effectiveness of learned augmentation, we have updated the following results (underlined) of 'Vanilla', 'Fixed', and ‘Random’ in Table 4 of the revised manuscript, by applying the extra loss terms. Here, DND still outperforms other baselines, and hence these results continuously demonstrate the superiority of learning augmentations via DND.
>
>  | Dataset  |  Vanilla  | Fixed  | Random | Difficult | Not Different | DND (no $w$) | DND |
> |:--------|:------------:|:----:|:------------:|:------------:|:------------:| :------------:|:------------:|
> | News20       | $\underline{82.49 \small{\pm 0.23}}$ | $\underline{84.06 \small{\pm 0.19}}$ | $\underline{84.58 \small{\pm 0.16}}$ |  $83.96 \small{\pm 0.12}$ |  $84.65 \small{\pm 0.13}$ |  $85.02 \small{\pm 0.20}$ |  $\bf 85.19 \small{\pm 0.35}$ |
> | Review50       | $\underline{71.58 \small{\pm 0.18}}$ | $\underline{73.57 \small{\pm 0.09}}$ | $\underline{73.04 \small{\pm 0.07}}$ |  $73.52 \small{\pm 0.06}$ |  $72.64 \small{\pm 0.02}$ |  $74.13 \small{\pm 0.16}$ |  $\bf 74.87 \small{\pm 0.17}$ |

---

> ### Author Response · Authors · 2021-11-20
> **Response to Reviewer XAnt (1/4)**
>
> We sincerely appreciate your efforts and insightful comments to improve the manuscript. We respond to each of your comments one-by-one in what follows. In the revised manuscript, we have marked the revisions with “green.” In tables, all the values and error bars are mean and standard deviation across 3 random seeds.
>
> ---
>
> **[C1] Comparison with other learning objectives**
>
> **[A1]** Following your suggestion, we additionally compare DND with another learning-based method, which directly maximizes the validation performance to update the augmentation policy. As auto-augment requires massive computations than the continuous relaxation method [Hataya et al., 2020], which is adopted to our method (e.g., 5000 vs. 0.23 GPU hours in CIFAR10 by [Hataya et al., 2020]), we instead consider LDM [Hu et al., 2019] that also maximizes the validation accuracy to learn the augmentation in a more efficient way, by adapting an off-the-shelf reward learning algorithm from RL. Since LDM originally uses a different augmentation pool, we modified LDM to use the same augmentation pool with DND. Also, we apply extra losses (Eq. 4 and Eq.5) with LDM as these losses slightly improve the overall performance, as shown in Appendix F of the revised draft.
>
>   As summarized below, we indeed observe that DND outperforms LDM. These results demonstrate that our reward design is more effective than maximizing the validation accuracy for learning the augmentations. Furthermore, we emphasize that maximizing validation accuracy usually requires more computations than training loss-based rewards due to an inherent bi-level optimization, as shown in [Zhang et al. 2020; Hataya et al., 2020]. For example, LDM is 3x slower than DND under the same update frequency in our experiments. We have added the discussion and the results of LDM in Section 4.3 and Table 4, respectively.
>
> | Dataset  |  LDM [Hu et al., 2019]  | DND |
> |:--------|:------------:|:----:|
> | News20       | $85.08 \small{\pm 0.07}$ | $\bf 85.19 \small{\pm 0.35}$ |
> | Review50    | $73.72 \small{\pm 0.22}$ | $\bf 74.87 \small{\pm 0.17}$ |
>
> ---
>
> **[C2] Sensitivity of DND hyperparameters**
>
> **[A2]** While we agree that DND has some hyper-parameters, DND is not sensitive to the choice of the hyper-parameters. To validate this, we additionally conduct the following experiments. Here, the original design choices of DND are underlined. In summary, these results indicate that DND is not sensitive to the choice of hyper-parameters. We have added all the respective discussions and results in Appendix G.
>
> - *Number of augmentation policies ($n_{\mathcal{T}}$)*: we compare the original choice ($n_{\mathcal{T}}=4$) with less ($n_{\mathcal{T}}=1$) and more ($n_{\mathcal{T}}=10$) number of policies. Overall, the improvement from DND is not sensitive to the number of augmentation policies, but using multiple policies shows slightly better results as it would construct more diverse augmentations.
>
> | Dataset  |  $n_{\tt policy}=1$  | $n_{\tt policy}=4$ | $n_{\tt policy}=10$  |
> |:--------|:------------:|:----:|:------------:|
> | News20       | $84.93 \small{\pm 0.05}$ | $\underline{85.19 \small{\pm 0.35}}$ | $85.16 \small{\pm 0.08}$ |
> | Review50    | $74.84 \small{\pm 0.18}$ | $\underline{74.87 \small{\pm 0.17}}$ | $74.90 \small{\pm 0.17}$ |
>
> - *Number of operations $T$*: we compare the original choice ($T=2$) with less ($T=1$) and more ($T=3,4$) number of operations. Here, one can verify that a single operation is not enough to construct complexly augmented samples, hence the empirical gain is relatively small compared to the original choice. With more operations, almost saturated improvement is observed. Remarkably, similar tendencies are also observed in vision tasks [Hataya et al., 2020; Cubuk et al., 2020].
>
> | Dataset  |  $T=1$  | $T=2$ | $T=3$  | $T=4$ |
> |:--------|:------------:|:----:|:------------:|:----:|
> | News20       | $84.68 \small{\pm 0.41}$ | $\underline{85.19 \small{\pm 0.35}}$ | $85.19 \small{\pm 0.25}$ | $85.22 \small{\pm 0.10}$ |
> | Review50    | $74.24 \small{\pm 0.07}$ | $\underline{74.87 \small{\pm 0.17}}$ | $74.93 \small{\pm 0.07}$ | $75.01 \small{\pm 0.02}$ |
>
> - *Significance of auxiliary operation*: here, we provide the results without an auxiliary operation [Hataya et al., 2020]. As one can see, this operation is not significant to the gain from DND.
>
> | Dataset  |   DND (without auxiliary operation)   | DND |
> |:--------|:------------:|:----:|
> | News20       | $85.14 \small{\pm 0.09}$ | $\underline{85.19 \small{\pm 0.35}}$ |
> | Review50    | $74.79 \small{\pm 0.07}$ | $\underline{74.87 \small{\pm 0.17}}$ |

---

### Author Response · Authors · 2021-11-20
**General Response**

Dear reviewers and AC,

We sincerely appreciate your valuable time and effort spent reviewing our manuscript. As reviewers highlighted, our work aims at an important and well-motivated (reviewer PytR, XAnt) problem, and provide a simple (PytR, SErr), yet intuitive (SErr) method that shows strong empirical results (PytR, SErr, XAnt) on the extensive experiments (all reviewers) with clear writing (PytR, A1ft, XAnt). In particular, we believe that DND makes a meaningful contribution to exploring the relatively under-explored area in the literature, as highlighted by reviewer PytR.

We appreciate your constructive feedback on our manuscript. In response to the comments, we have carefully revised and enhanced the manuscript, including the following additional discussions and experiments:
- Adding caution on the used terminology (Section 3.1)
- More clarification of the method description (Section 3.2)
- More detailed description for used data augmentation pool (Section 4.1 and Appendix A.3)
- Updating experimental results of some baselines with more hyper-parameter tuning (Figure 1 and Tables 1-6)
- Conducting the statistical significance test (Section 4.2)
- Applying the proposed losses and adding the comparison with another learning-based method (Table 4)
- Move paragraph “Difficult, but not different samples” in Section 4.3 into Appendix D for the improved organization
- Additional discussions on the future directions (Section 5)
- Additional discussions and experimental results on the extra losses (Figures 8-9, Tables 8-10, and Appendix F)
- More ablation studies on the designs of DND (Tables 11-17 and Appendix G)

In the revised manuscript, these updates are temporarily highlighted in "green” for your convenience to check.

We sincerely believe that these updates may help us better deliver the benefits of the proposed algorithm DND to the ICLR community.

Thank you very much,

Authors.

---

### Public Comment · ~Lei_Li14 · 2022-03-02
**Relevant paper regarding learning augmentation policy**

Thanks for your inspiring work towards learning augmentation policy for NLP tasks, and finding that DND samples are vital and the proposed re-weighting schema are quite interesting.

We have previously investigated text auto-augmentation in [1] by designing parameterized learning policy and optimizing it via an efficienct Bayesian optimization algorithm, which we think can be added to the **Learning-based data augmentation** section for a more comprehensive introduction of current progress.

[1] [Text AutoAugment: Learning Compositional Augmentation Policy for Text Classification](https://aclanthology.org/2021.emnlp-main.711) (Ren et al., EMNLP 2021)

---

### Decision · Program_Chairs · 2022-01-20

**Decision:**

Accept (Poster)

**Comment:**

We appreciate the authors for addressing the comments raised by the reviewers during the discussion period, which includes providing more experimental results to address the concerns. We believe the publication of this paper can contribute to the important topic of data augmentation.

The authors are highly recommended to consider all the comments and suggestions made by the reviewers when further revising their paper for publication.